# UGC: Universal Graph Coarsening

**Mohit Kataria**[1]
Mohit.Kataria@scai.iitd.ac.in

**Sandeep Kumar**[2,1,3]
ksandeep@ee.iitd.ac.in

**Jayadeva**[2,1]
jayadeva@ee.iitd.ac.in

[1] Yardi School of Artificial Intelligence
[2]Department of Electrical Engineering
[3]Bharti School of Telecommunication Technology and Management
Indian Institute of Technology Delhi

## Abstract

In the era of big data, graphs have emerged as a natural representation of intricate relationships. However, graph sizes often become unwieldy, leading to storage, computation, and analysis challenges. A crucial demand arises for methods that can effectively downsize large graphs while retaining vital insights. Graph coarsening seeks to simplify large graphs while maintaining the basic statistics of the graphs, such as spectral properties and $\epsilon$-similarity in the coarsened graph. This ensures that downstream processes are more efficient and effective. Most published methods are suitable for homophilic datasets, limiting their universal use. We propose **U**niversal **G**raph **C**oarsening (UGC), a framework equally suitable for homophilic and heterophilic datasets. UGC integrates node attributes and adjacency information, leveraging the dataset's heterophily factor. Results on benchmark datasets demonstrate that UGC preserves spectral similarity while coarsening. In comparison to existing methods, UGC is $4\times$ to $15\times$ faster, has lower eigen-error, and yields superior performance on downstream processing tasks even at 70% coarsening ratios.[1]

## 1 Introduction

Graphs have emerged as highly expressive tools to represent diverse structures and knowledge in various fields such as social networks, bio-informatics, transportation, and natural language processing [1–3]. They are essential for tasks like community detection, drug discovery, route optimization, and text analysis. With the growing importance of graph-based solutions, dealing with large graphs has become a challenge. **G**raph **C**oarsening(GC), a widely used technique to simplify graphs while retaining vital information, making them more manageable for analysis [4]. It has been applied successfully in various tasks [5–10]. Preserving the structural information of the graph is crucial in graph coarsening algorithms to ensure the fidelity of the coarsened graphs. A high-quality coarsened graph retains essential features and relationships, enabling accurate results for downstream tasks. Additionally, computational efficiency is equally vital for scalability, as large-scale graphs are common in real-world applications. An efficient coarsening method should ensure that the reduction in graph size does not come at the expense of excessive computation time but existing graph coarsening methods often face trade-offs between scalability and the quality of the coarsened graph. Our method draws inspiration from hashing techniques, which provide us with advantages in terms of computational efficiency. As a result, our approach exhibits a linear time complexity, making it highly efficient even for large graphs.

Graph datasets often exhibit a blend of homophilic and heterophilic traits [11, 12]. **G**raph **C**oarsening(GC) has been widely explored on homophilic datasets, but, to the best of our knowledge, has never been applied to heterophilic graphs. We propose Universal Graph Coarsening $UGC$, an approach that works well on both. Figure 2 illustrates how UGC uses a graph's adjacency matrix as

---

[1]Code is available at [UGC](UGC)

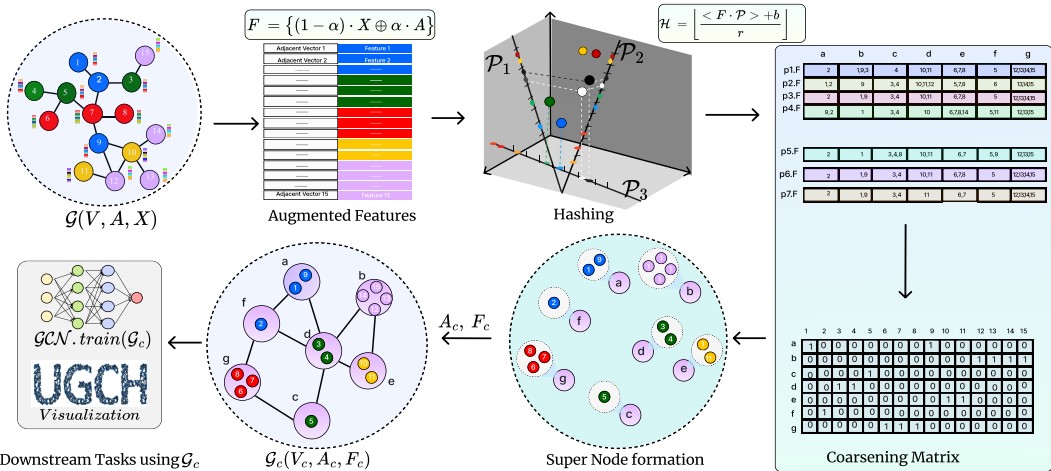

Figure 2: This figure illustrates our framework, UGC, which has three main modules a) Generation of an augmented matrix by incorporating feature and adjacency matrices while using heterophily measure $\alpha$, b) Generation of coarsening matrix $\mathcal{C}$ using augmented features via Hashing, and c) Generation of coarsened graph $\mathcal{G}_c$ from $\mathcal{C}$ followed by its utilization in downstream tasks.

well as the node feature matrix. UGC relies on hashing, lending computational efficiency. UGC exhibits linear time complexity, enabling fast processing of large datasets. Figure 1 demonstrates the computational time gains of UGC among graph coarsening methods. UGC surpasses the fastest existing methods by about $6\times$ on the Physics dataset and $9\times$ on the Squirrel dataset. UGC enhances the performance of Graph Neural Networks (GNN) models in classification tasks, indicating its suitability for downstream processing. UGC coarsened graphs retain essential spectral properties and show low eigen error, hyperbolic error, and $\epsilon$-similarity measure. In a nutshell, UGC is fast, universally applicable, and information-preserving.

**Key Contributions.**

- We proposed a novel framework that is extremely fast compared to other existing methods for graph coarsening. It is also shown to be helpful and effective for graph-based downstream tasks.
- UGC is the first to handle heterophilic datasets for coarsening.
- UGC can retain important spectral properties, such as eigen error, hyperbolic error, and $\epsilon$-similarity measure, which ensures the preservation of key characteristics and information of the original graph during the graph coarsening.

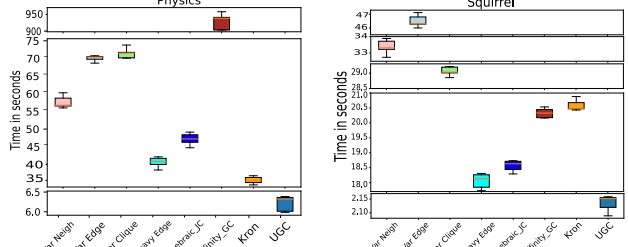

Figure 1: This figure illustrates the computational time comparison among graph coarsening methods to learn a coarsened graph over ten iterations. UGC outperforms the fastest existing methods by approximately $6\times$ on the Physics dataset and $9\times$ on the Squirrel dataset.

## 2 Background and Problem Formulation

A graph is represented using $\mathcal{G}(V, A, X)$ where $V = \{v_1, \cdots, v_N\}$ denotes set of $N$ vertices, $A \in \mathbb{R}^{N \times N}$ is the adjacency matrix and $A_{ij} > 0$ indicates an edge $(v_i, v_j)$ between nodes $v_i$ and $v_j$. $X \in \mathbb{R}^{N \times \widetilde{d}}$ denotes the feature matrix where $i^{th}$ row of $X$ is a feature vector $X_i \in \mathbb{R}^{\widetilde{d}}$, associated with node $v_i$. The degree matrix $D$ is a diagonal matrix, where $D_{ii} = \sum_j A_{ij}$. $L \in \mathbb{R}^{N \times N}$ is a Laplacian matrix, $L = D - A$ [13], and it belongs to the set $S_L = \{L \in \mathbb{R}^{N \times N} | L_{ji} = L_{ij} \leq 0, \; \forall i \neq j; \; L_{ii} = -\sum_{j \neq i} L_{ij}\}$ as defined in [14, 15]. The adjacency matrix $A$ and Laplacian matrix $L$ associated with the graph are related as follows: $A_{ij} = -L_{ij}$ for $i \neq j$ and $A_{ij} = 0$ for $i = j$.

$$\mathcal{C}^T = \begin{bmatrix} 1 & 1 & 1 & 1 & 0 & 0 & 0 & 0 \\ 0 & 0 & 0 & 0 & 1 & 0 & 0 & 0 \\ 0 & 0 & 0 & 0 & 0 & 1 & 1 & 1 \end{bmatrix}$$

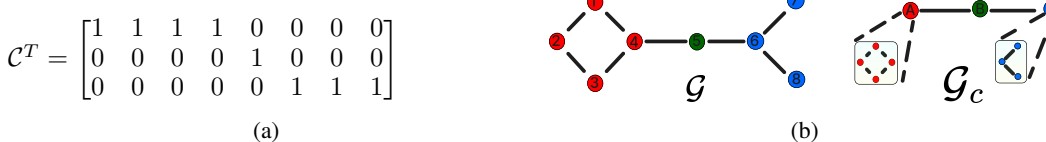

(a)                                    (b)

Figure 3: Graph coarsening toy example, a) Coarsening matrix, b) Original graph $\mathcal{G}$ and corresponding coarsened graph $\mathcal{G}_c$

Both $L$ and $A$ can represent the same graph. Hence, a graph $\mathcal{G}(V, A, X)$ can also be represented as $\mathcal{G}(L, X)$, with either representation utilized as required within the paper.

**Problem.** The objective is to reduce an input graph $\mathcal{G}(V, A, X)$ with $N$-nodes into a new graph $\mathcal{G}_c(\widetilde{V}, \widetilde{A}, \widetilde{X})$, with $n$-nodes and $\widetilde{X} \in \mathbb{R}^{n \times \widetilde{d}}$ where $n \ll N$. The **G**raph **C**oarsening(GC) problem requires learning of a coarsening matrix $\mathcal{C} \in \mathbb{R}^{N \times n}$, which is a linear mapping from $V \to \widetilde{V}$. A linear mapping ensures that similar nodes in $\mathcal{G}$ are mapped to the same super-node in $\mathcal{G}_c$, s.t. $\widetilde{X} = \mathcal{C}^T X$. Every non-zero entry $\mathcal{C}_{ij}$ denotes the mapping of the $i^{th}$ node of $\mathcal{G}$ to the $j^{th}$ super-node $\mathcal{G}_c$. This $\mathcal{C}$ matrix belongs to the following set:

$$\mathcal{S} = \left\{ \mathcal{C} \in \mathbb{R}^{N \times n}, \mathcal{C}_{ij} \in \{0, 1\}, \|\mathcal{C}_i\| = 1, \langle \mathcal{C}_i^T, \mathcal{C}_j^T \rangle = 0, \forall i \neq j, \langle \mathcal{C}_l, \mathcal{C}_l \rangle = d_{\widetilde{V}_l}, \|\mathcal{C}_i^T\|_0 \geq 1 \right\} \quad (1)$$

where $d_{\widetilde{V}_l}$ means the number of nodes in the $l^{th}$-supernode. The condition $\langle \mathcal{C}_i^T, \mathcal{C}_j^T \rangle = 0$ ensures that each node of $\mathcal{G}$ is mapped to a unique super-node. The constraint $\|\mathcal{C}_i^T\|_0 \geq 1$ requires that each super-node contains at least one node. Consider the 8-node graph in Figure 3b. Nodes 1, 2, 3, and 4 are mapped to super-node **A**, while nodes 6, 7, and 8 are mapped to super-node **C**. Hence, the coarsening matrix $\mathcal{C}$ is given in Figure 3a. The goal is to learn this $\mathcal{C}$ matrix such that $\mathcal{G}$ and $\mathcal{G}_c$ are similar. The $\epsilon-$similarity is a widely used similarity measure for graphs with node features, as it entails comparing the Laplacian norms of the respective feature matrices. The graphs $\mathcal{G}(V, A, X)$ and $\mathcal{G}_c(\widetilde{V}, \widetilde{A}, \widetilde{X})$ are said to be $\epsilon$-similar if there exist $\epsilon \geq 0$ such that

$$(1 - \epsilon)\|X\|_L \leq \|\widetilde{X}\|_{L_c} \leq (1 + \epsilon)\|X\|_L \quad (2)$$

where $L$ and $L_c$ are the Laplacian matrices of $\mathcal{G}$ and $\mathcal{G}_c$ respectively, $\|X\|_L = \sqrt{tr(X^T L X)}$ and $\|\widetilde{X}\|_{L_c} = \sqrt{tr(\widetilde{X}^T L_c \widetilde{X})}$. The quantity $tr(X^T L X) = -\sum_{i,j} L_{ij} \|x_i - x_j\|^2$ is known as Dirichlet Energy (DE), which is employed to measure the smoothness of node features where $x_i$ and $x_j$ are the node features of nodes $i$ and $j$ [14].

***Goal**: Given a graph $\mathcal{G}(V, A, X)$ of $N$ nodes, construct a coarsened graph $\mathcal{G}_c(\widetilde{V}, \widetilde{A}, \widetilde{X})$ with $n$ nodes, such that they are $\epsilon-$similar.*

**Homophilic and Heterophilic datasets.** Graph datasets may demonstrate homophily and heterophily properties [16–19]. Homophily refers to the tendency of nodes to be connected to other nodes of the same class or type, while heterophily signifies the tendency of nodes to connect with nodes of different classes. A heterophily factor $0 \leq \alpha \leq 1$ may be used to denote the degree of heterophily. $\alpha$ is calculated as the fraction of edges between nodes of different classes to the total number of edges. A strongly heterophilic graph ($\alpha \to 1$) has the most edges between nodes of different classes, suggesting a diverse network with mixed interactions. Conversely, weak heterophily or strong homophily ($\alpha \to 0$) occurs in networks where nodes predominantly connect with others of the same class.

**Locality Sensitive Hashing.** Locality Sensitive Hashing (LSH) is a linear time, efficient similarity search technique for high dimensional data [20–23]. It maps high-dimensional vectors to lower dimensions while ensuring that similar vectors collide with high probability. LSH uses a family of hash functions to map vectors to buckets, enabling fast retrieval and similarity search. It has found applications in image retrieval [24], data mining [25], and similarity search algorithms [26]. LSH family is defined as

**Definition 2.1** *Let $d$ be a distance measure, and let $d_1 < d_2$ be two distances. A family of functions $F$ is said to be $(d_1, d_2, p_1, p_2)-$sensitive if for every $f \in F$ the following two conditions hold:*

*1. If $d(x, y) \leq d_1$ then probability $[f(x) = f(y)] \geq p_1$*
*2. If $d(x, y) \geq d_2$ then probability $[f(x) = f(y)] \leq p_2$*

UGC uses LSH with a set of random projectors to map similar nodes to the same super-node. The projection is computed as $\left\lfloor \frac{<x \cdot w_i> + b_i}{r} \right\rfloor$, where $w_i$ is a randomly selected $d-$dimensional projector vector from a $p-$stable distribution (see Appendix A); $x$ represents the $d-$dimensional data sample, and $r$ is the width of each quantization bin.

**Related Works.** The literature is replete with graph reduction methods and their applications; they may be broadly classified into three categories:

1. *Optimization and Heuristics:* Loukas [15] proposed advanced spectral graph coarsening algorithms based on local variation to preserve the original graph's spectral properties. Two variants, *viz.* edge-based (LVE) and neighborhood-based (LVN), select contraction sets with small local variation in each stage but have limitations in achieving arbitrary coarsening levels. Heavy edge matching (HE) [9, 27], determines the contraction family by computing a maximum-weight matching based on the weight of each contraction set. The Algebraic Distance method proposed in [27, 28] calculates the weight of each candidate set using an algebraic distance measure. The affinity method [29], inspired by algebraic distance, uses the vertex proximity heuristic. The Kron reduction method [30] was originally proposed for electrical networks but is too slow for large networks. FGC [14, 31] considers both the graph structure and the node attributes as the input and, alternatively, optimizes $\mathcal{C}$. The above-mentioned methods are computationally and memory-intensive.

2. *GNN based:* GCond [32] and SFGC [33] are GNN-based graph condensation methods. These works proposed the online gradient matching schema between the synthesized small-scale graph and the large-scale graph. However, these methods have significant issues regarding computational time and generalization ability. First, they require training GNN models on the original graph to get a smaller graph as they imitate the GNN training trajectory on the original graph through gradient matching. Due to this, these methods are extremely computationally demanding and may not be suitable for the scalability of GNN models. However, these methods can be beneficial for other tasks, like solving storage and visualization issues. Second, the condensed graph obtained using GCond [32] shows poor generalization ability across different GNN models [33] because different GNN models vary in their convolution operations along graph structures.

3. *Scaling GNN viz. Graph Coarsening:* SCAL [34] and GOREN [35] proposed to enhance the scalability for training GNN models using graph coarsening. It is worth noting that SCAL and GOREN are not standalone graph coarsening techniques. SCAL uses Louka's [15] work to coarsen the graph, then trains GNN models using the coarsened graph. While GOREN trying to improve the coarsening quality of existing methods.

## 3 The Proposed Framework: Universal Graph Coarsening (UGC)

The proposed UGC framework comprises three main components: (a) First, obtaining an augmented feature matrix $F$ containing both node feature and structural information, (b) Secondly, using locality-sensitive hashing to derive the coarsening matrix $\mathcal{C}$, (c) and Finally, obtaining the coarsened graph adjacency matrix $A_c$ and coarsened features $F_c$.

**Construction of Augmented Feature Matrix $F$.** In order to create a universal GC framework suitable for all, it is important to consider features at both i) the node level, i.e., features, and ii) the structure-level, i.e., adjacency matrix, together. In this regard, we create an augmented feature matrix $F$, where each node's feature vector $X_i$ is augmented with its binary adjacency vector $A_i$. We use the heterophily factor $\alpha$ discussed in Section 2 to balance the emphasis between node-level and structure-level information. The augmented feature vector for node $v_i$ is given using $F_i = \left\{ (1-\alpha) \cdot X_i \oplus \alpha \cdot A_i \right\}$ where $\oplus$ and $\cdot$ denote the concatenation and dot product operations, respectively. Figure 11 in Appendix K illustrates a toy example of the process involved in calculating the augmented feature vector. While larger graphs may result in long vectors, efficient implementations and sparse tensor methods may alleviate this hurdle. A motivating example demonstrating the need for augmented features while doing GC is discussed in Appendix K (Figure 12).

**Construction of Coarsening Matrix $\mathcal{C}$.** Let $F_i \in \mathbb{R}^d$ represent the augmented feature vector of node $v_i$. Let $\mathcal{W} \in \mathbb{R}^{d \times l}$ and $b \in \mathbb{R}^l$ be the hashing matrices used in UGC, with $l$ denoting the number of hash functions. The hash indices generated by $k^{th}$ hash/projector function for $i^{th}$ node is given as

$$h_i^k = \left\lfloor \frac{1}{r} * (\mathcal{W}_k \cdot F_i + b_k) \right\rfloor \tag{3}$$

where $r$ is a hyperparameter called bin-width. The hash index that has the maximum occurrence among the hash indices generated by all $l$ hash functions is the hash value assigned to the graph node $v_i$. Hence, the hash value for node $v_i$ is given by

$$h_i = maxOccured\{h_i^1, h_i^2....h_i^l\} \tag{4}$$

$r$ controls the size of the coarsened graph $\mathcal{G}_c$; empirically, we find that increasing $r$ means reducing the size of the coarsened graph $\mathcal{G}_c$. All nodes assigned with the same hash value map to the same super-node in $\mathcal{G}_c$. The reader may like to refer to Algorithm 1 for the steps in UGC. The element of coarsening matrix, $\mathcal{C}_{ij}$ equals 1 if vertex $v_i$ is associated with super-node $\widetilde{v}_j$. Crucially, every node is assigned a unique $h_i$ value, ensuring an exclusive mapping to a super-node. This constraint aligns with the formulation of super-node and guarantees at least one node per super-node. Thus, each row of $\mathcal{C}$ contains only one non-zero entry, leading to orthogonal columns. This matrix $\mathcal{C}$ satisfies the conditions specified in Equation 1.

---

**Algorithm 1** UGC: Universal Graph Coarsening

---

**Require:** Input $\mathcal{G}(V, A, X)$, $l \leftarrow$ Number of Projectors, $r \leftarrow binWidth$
1: $\alpha = \frac{|\{(v,u) \in E: y_v = y_u\}|}{|E|}$; $\alpha$ is heterophily factor, $y_i \in \mathbb{R}^N$ is node labels, $E$ denotes edge list
2: $F = \{(1 - \alpha) \cdot X \oplus \alpha \cdot A\}$
3: $\mathcal{W} \sim \mathcal{D}(.)$; $\mathcal{W} \in \mathbb{R}^{d \times l}$ denotes $l$ projectors, and $\mathcal{D}$ is a p-stable distribution
4: $b \sim \mathcal{D}(.)$; $b \in \mathbb{R}^l$ denotes sampled bias
5: $\mathcal{H} = \lfloor \frac{<F \cdot \mathcal{W}> + b}{r} \rfloor$; $\mathcal{H} \in \mathbb{R}^{N \times l}$
6: $\pi_i \leftarrow maxOccurence(\mathcal{H}_i; i \in \{1, 2, 3, ..., N\})$, $\pi \in \mathbb{R}^N$
7: **for** every node v in V **do**
8: $\quad \mathcal{C}[v, \pi[v]] \leftarrow 1$
9: $A_c(i, j) \leftarrow \sum_{(u \in \pi^{-1}(\widetilde{v}_i), v \in \pi^{-1}(\widetilde{v}_j))} A_{uv}, \forall i, j \in \{1, 2, ..., n\}$
10: $F_c(i) \leftarrow \frac{1}{|\pi^{-1}(\widetilde{v}_i)|} \sum_{u \in \pi^{-1}(\widetilde{v}_i)} F_u, \forall i \in \{1, 2, ..., n\}$
11: return $\mathcal{G}_c(V_c, A_c, F_c), \mathcal{C}$

---

**Construction of Coarsened Graph $\mathcal{G}_c$.** Let $\mathcal{G}_c(\widetilde{V}, \widetilde{A}, \widetilde{F})$ represent the coarsened graph that is to be built. A pair of super-nodes, say $\widetilde{v}_i$ and $\widetilde{v}_j$, in the coarsened graph $\mathcal{G}_c$ are connected, if any of the nodes, say $u \in \pi^{-1}(\widetilde{v}_i)$ has an edge to any of the nodes, say $v \in \pi^{-1}(\widetilde{v}_j)$ in the original graph, i.e., $\exists\, u \in \pi^{-1}(\widetilde{v}_i), v \in \pi^{-1}(\widetilde{v}_j)$ such that $A_{uv} \neq 0$. The coarsened graph $\mathcal{G}_c$ is weighted, and the weight assigned to the edge between nodes $\widetilde{v}_i$ and $\widetilde{v}_j$, is given by $\widetilde{A}_{ij} = \sum_{(u \in \pi^{-1}(\widetilde{v}_i), v \in \pi^{-1}(\widetilde{v}_j))} A_{uv}$ where $A_{uv}$ refers to the element $(u, v)$ in the adjacency matrix $A$ of graph $\mathcal{G}$. The features of super-nodes are taken to be the average of the features of the nodes in the super-node, i.e., $\widetilde{F}_i = \frac{1}{|\pi^{-1}(\widetilde{v}_i)|} \sum_{u \in \pi^{-1}(\widetilde{v}_i)} F_u$. The super-node's label is chosen as the class that has the most instances. From the $\mathcal{C}$ matrix, we can directly calculate the adjacency $\widetilde{A}$ matrix of $\mathcal{G}_c$ using $\widetilde{A} = \mathcal{C}^T A \mathcal{C}$ which is the same as $\widetilde{A}_{ij}$. $\widetilde{F}$ can also be obtained using $\widetilde{F} = \mathcal{C}^T F$ where $\mathcal{C}$ is the coarsening matrix discussed earlier. Because each super-edge combines multiple edges from the original graph, the number of edges in the coarse graph is also much less than $m$. In general, the adjacency matrix $\widetilde{A}$ has a substantially smaller number of non-zero elements than $A$. The pseudocode for UGC is listed in Algorithm 1. UGC gives a coarsened graph $\mathcal{G}_c(L_c, \widetilde{F})$ which also satisfies $\epsilon-$similarity ($\epsilon \geq 0$).

**Theorem 3.1** *The input graph $\mathcal{G}(L, F)$ and the coarsened graph $\mathcal{G}_c(L_c, \widetilde{F})$ obtained using the proposed UGC algorithm are $\epsilon$-similar with $\epsilon \geq 0$, i.e.,*

$$(1 - \epsilon)\|F\|_L \leq \|\widetilde{F}\|_{L_c} \leq (1 + \epsilon)\|F\|_L \tag{5}$$

*where $L$ and $L_c$ are the laplacian matrices of $\mathcal{G}$ and $\mathcal{G}_c$ respectively.*

***Proof:*** The proof is deferred in Appendix I.

**Universal Graph Coarsening with feature re-learning for Bounded $\epsilon$-similarity.** The coarsened graph $\mathcal{G}_c$ generated through UGC exhibits a high degree of similarity, within the range of $\epsilon$, to the original graph $\mathcal{G}$. It has also been empirically demonstrated that this coarsened representation performs exceptionally well across various downstream tasks. Nonetheless, to achieve a tighter $\epsilon$-bound, where ($\epsilon \leq 1$), a potential step involves introducing modifications to the feature learning procedure of the super-nodes $\mathcal{G}_c$.

It is important to note that the $\epsilon$-similarity measure introduced in [15] does not incorporate features. Instead, it relies on the eigenvector of the laplacian matrix to compute similarity, which limits its ability to capture the characteristics of the associated features along with the graph structure. Once we get the loading matrix $\mathcal{C}$ using UGC as discussed in Section 3 we used $\widetilde{F}_i = \frac{1}{|\pi^{-1}(\widetilde{v}_i)|} \sum_{u \in \pi^{-1}(\widetilde{v}_i)} F_u$ to learn the feature-vectors of super-nodes. Using $\widetilde{F}_i$ we can satisfy the Theorem 3.1. However, to give a strict bound on the $\epsilon$ similarity we updated $\widetilde{F}$ to $\widehat{F}$ by minimizing the term

$$\min_{\widehat{F}} f(\widehat{F}) = \text{tr}(\widehat{F}^T \mathcal{C}^T L \mathcal{C} \widehat{F}) + \frac{\alpha}{2} \|\mathcal{C}\widehat{F} - F\|_F^2 \tag{6}$$

We aim to enforce the Dirichlet smoothness condition in super-node features using Equation 6. The above equation is a convex optimization problem from which we get a closed-form solution by putting the gradient w.r.t to $\widehat{F}$ equal to zero. Update rule for $\widehat{F}$ can be derived as:

$$2\mathcal{C}^T L \mathcal{C} \widehat{F} + \alpha \mathcal{C}^T (\mathcal{C}\widehat{F} - F) = 0 \implies \widehat{F} = \left( \frac{2}{\alpha} \mathcal{C}^T L \mathcal{C} + \mathcal{C}^T \mathcal{C} \right)^{-1} \mathcal{C}^T F$$

We now have re-learnt features for super-nodes, please refer to Algorithm 2 in Appendix B which we call as **UGC-FL** i,e UGC with feature learning. Using $\widehat{F}$ we can give a more strict bound on $\epsilon-$similarity.

**Theorem 3.2** *The original graph* $\mathcal{G}(L, F)$ *and coarsened graph* $\mathcal{G}_c(L_c, \widehat{F})$ *obtained using the proposed UGC-FL algorithm are $\epsilon$ similar with $0 < \epsilon \leq 1$, i.e,*

$$(1 - \epsilon)\|F\|_L \leq \|\widehat{F}\|_{L_c} \leq (1 + \epsilon)\|F\|_L \tag{7}$$

*where $L$ and $L_c$ are the laplacian matrices of $\mathcal{G}$ and $\mathcal{G}_c$ respectively, and $F$ and $\widehat{F}$ are features matrix associated with original and coarsened graphs, respectively.*

***Proof:*** The proof is deferred in Appendix J.

***Novelty:*** The majority of current techniques involve coarsening the original graph and simultaneously learning the graph structure, which makes them computationally intensive. The UGC decouples this process, making it incredibly fast, first learning the coarsening mapping $C$ by capturing the similarity of features through hashing and then using the adjacency matrix only once as $A_c = C^T A C$ for learning the coarsened graph's structure all at once. The UGC is easy to use, extremely fast, and produces better results for tasks requiring downstream processing.

**Time Complexity Analysis of UGC.** We have three phases for our framework. For the first phase, we can see Algorithm 1, Line 5 is driving the complexity of the algorithm, where we multiply two $F \in \mathbb{R}^{N \times d}$ and $\mathcal{W} \in \mathbb{R}^{d \times l}$ matrices, which results in $\mathcal{O}(Nld)$. In the second pass, the super-nodes for the coarsened graphs are constructed with the help of the accumulation of nodes in the bins. The main contribution of UGC is up to these two phases i.e., Line 1-8. Till now, time-complexity is $\mathcal{O}(Nld) \equiv \mathcal{O}(NC)$ where $C$ is a constant.

In the third phase, Lines 10-11, we calculate the adjacency and features of the super-nodes of the coarsened graph $\mathcal{G}_c$. The computational cost of this operation is $\mathcal{O}(m)$, where $m$ is the number of edges in the original graph $\mathcal{G}$, and this is a one-time step. Indeed, the overall time complexity of all three phases combined is $\mathcal{O}(N + m)$ where $m$ is the number of edges. However, it's important to note that the primary contribution of UGC lies in the process of finding the coarsening matrix, whose time complexity is $\mathcal{O}(N)$. We have compared the computational time for obtaining the coarsening matrix via UGC with the existing methods.

## 4 Experiments

In this section, we conduct extensive experiments to evaluate the proposed UGC against the existing graph coarsening algorithms. The conducted experiments establish the performance of UGC concerning i) computational efficiency, ii) preservation of spectral properties, and iii) potential extensions of the coarsened graph $\mathcal{G}_c$ into real-world applications.

We compare our proposed algorithm with the following coarsening algorithms, as discussed in Section 2. UGC (feat) represents a specific scenario within our framework, wherein only the feature values are considered for hashing, thereby obtaining the mapping of super-nodes. To comprehend the

significance of incorporating the adjacency vector, we have added the results for both UGC (feat) and UGC (augmented feat).

**Datasets.** Our experiments cover widely adopted benchmarks, including *Cora ,Citeseer, Pubmed* [36], *CS, Physics* [37], *DBLP* [38]. Additionally, UGC effectively coarsens large datasets like *Flickr, Yelp* [39], and *Reddit* [40], previously challenging for existing techniques. We also present datasets like *Squirrel, Chameleon, Texas, Film, Wisconsin* [11, 12, 16, 17], characterized by dominant heterophilic factors. Table 6 in Appendix G provides comprehensive dataset details.

Table 1: Summary of run-time in seconds averaged over 5 runs to reduce the graph to 50%.

| Data/Method | Cora | Cite. | CS | PubMed | DBLP | Physics | Flickr | Reddit | Yelp | Squirrel | Cham. | Cor. | Texas | Film |
|---|---|---|---|---|---|---|---|---|---|---|---|---|---|---|
| Var. Neigh. | 6.64 | 8.72 | 23.43 | 24.38 | 22.79 | 58.0 | OOM | OOM | OOM | 33.26 | 12.2 | 1.34 | 0.63 | 27.67 |
| Var. Edges | 5.34 | 7.37 | 16.72 | 18.69 | 20.59 | 67.16 | OOM | OOM | OOM | 46.45 | 12.65 | 1.31 | 0.76 | 26.6 |
| Var. Cliq. | 7.29 | 9.8 | 24.59 | 61.85 | 38.31 | 69.80 | OOM | OOM | OOM | 28.91 | 10.55 | 1.56 | 1.14 | 33.04 |
| Heavy Edge | 0.7 | 1.41 | 7.50 | 12.03 | 8.39 | 39.77 | OOM | OOM | OOM | 18.08 | 5.41 | 1.62 | 1.17 | 11.79 |
| Alg. Dist | 0.93 | 1.55 | 9.63 | 10.48 | 9.67 | 46.42 | OOM | OOM | OOM | 18.03 | 5.24 | 1.58 | 0.81 | 12.65 |
| Affinity GS | 2.36 | 2.53 | 169.05 | 168.3 | 110.9 | 924.7 | OOM | OOM | OOM | 20.00 | 5.83 | 1.81 | 1.24 | 20.65 |
| Kron | 0.63 | 1.37 | 8.72 | 5.81 | 7.09 | 34.53 | OOM | OOM | OOM | 20.62 | 7.25 | 1.73 | 0.97 | 12.29 |
| UGC | **0.41** | **0.71** | **3.1** | **1.62** | **1.86** | **6.4** | **8.9** | **16.17** | **170.91** | **2.14** | **0.49** | **0.04** | **0.03** | **1.38** |

**Run-Time Analysis.** UGC's main contribution lies in its computational efficiency. The time required to compute the coarsening matrix $\mathcal{C}$ is summarized in Table 1. By referring to this Table, it becomes evident that UGC exhibits a remarkable advantage, surpassing all existing methods across diverse datasets. Our model outperforms existing methods by a substantial margin. While other methods struggle at large datasets like *Physics(34.4k nodes)*, UGC is able to coarsen down massive datasets like *Yelp(716.8k nodes)*, which was previously not possible. It should be emphasized that the time taken by UGC on the *Reddit(232.9k nodes)* dataset, which has $7\times$ the number of nodes compared to *Physics* is one-third the time taken by the fastest existing methods on *Physics* dataset.

**Spectral Properties Preservation.**

1. **Relative Eigen Error (REE):**, REE used in [14, 15, 41] gives the means to quantify the measure of the eigen properties of the original graph $\mathcal{G}$ that are preserved in coarsened graph $\mathcal{G}_c$.

   **Definition 4.1** *REE is defined as follows:* $REE(L, L_c, k) = \frac{1}{k} \sum_{i=1}^{k} \frac{|\widetilde{\lambda}_i - \lambda_i|}{\lambda_i}$ *where $\lambda_i$ and $\widetilde{\lambda}_i$ are top $k$ eigenvalues of original graph Laplacian ($L$) and coarsened graph Laplacian ($L_c$) matrix, respectively.*

2. **Hyperbolic error (HE):** HE [42] indicates the structural similarity between $\mathcal{G}$ and $\mathcal{G}_c$ with the help of a lifted matrix along with the feature matrix $X$ of the original graph.

   **Definition 4.2** *HE is defined as follows:* $HE = arccosh(\frac{||(L - L_{\text{lift}})X||_F^2 ||X||_F^2}{2 trace(X^T LX) trace(X^T L_{\text{lift}} X)} + 1)$ *where $L$ is the Laplacian matrix and $X \in \mathbb{R}^{N \times d}$ is the feature matrix of the original input graph, $L_{\text{lift}}$ is the lifted Laplacian matrix defined in [41] as $L_{\text{lift}} = \mathcal{C} L_c \mathcal{C}^T$ where $\mathcal{C} \in \mathbb{R}^{N \times n}$ is the coarsening matrix and $L_c$ is the Laplacian of $\mathcal{G}_c$.*

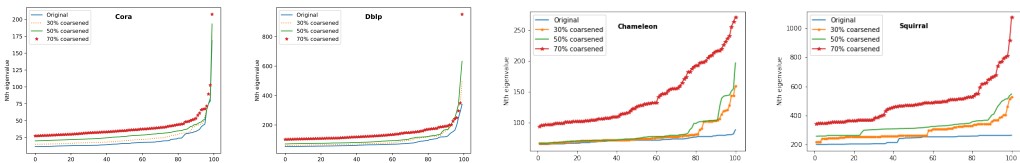

Figure 4: Top 100 eigenvalues of the original graph $\mathcal{G}$ and coarsened graph $\mathcal{G}_c$ at different coarsening ratios: 30%, 50%, and 70%.

Eigenvalue preservation can be seen in Figure 4 where we have plotted the top 100 eigenvalues of $\mathcal{G}$ and of $\mathcal{G}_c$. We can see that the spectral property is preserved even for 70% coarsened graphs. This approximation is more accurate for a lower coarsening ratio, i.e., the smaller the graph, the bigger the REE. The REE for all approaches across all datasets is shown in Table 2 for a fixed 50% coarsening ratio. UGC stands out by giving the best REE values in 8 out of 12 datasets. Although we also have coarsened graphs for large datasets like *Yelp and Reddit*, eigen error calculation for these datasets was out of memory, so we have used EOOM while other methods fail to find even the coarsened

Table 2: This table illustrates Relative Eigen Error at 50% coarsening ratio. UGC stands out by giving the best REE values in 8 out of 12 datasets.

| Data/Method | Cora | Cite. | CS | PubMed | DBLP | Physics | Flickr | Reddit | Yelp | Squirrel | Cham. | Cor. | Texas | Film |
|---|---|---|---|---|---|---|---|---|---|---|---|---|---|---|
| Var. Neigh. | 0.121 | 0.180 | 0.248 | 0.108 | 0.117 | 0.273 | OOM | OOM | OOM | 0.871 | 0.657 | 0.501 | 0.391 | 32.87 |
| Var. Edges | 0.129 | 0.136 | 0.049 | 0.965 | 0.135 | 0.042 | OOM | OOM | OOM | 0.298 | 0.597 | 0.485 | 0.489 | 21.8 |
| Var. Cli. | 0.085 | 0.064 | **0.026** | 1.208 | 0.082 | 0.039 | OOM | OOM | OOM | 0.369 | 0.456 | 0.550 | 0.463 | 22.95 |
| Hea. Edge | 0.071 | 0.043 | 0.046 | 0.834 | 0.086 | 0.031 | OOM | OOM | OOM | 0.256 | 0.333 | 0.554 | 0.464 | 5.69 |
| Alg. Dist. | 0.107 | 0.111 | 0.087 | 0.403 | 0.047 | 0.117 | OOM | OOM | OOM | 0.245 | 0.413 | 0.552 | 0.465 | 5.71 |
| Aff. GS | 0.095 | 0.057 | 0.063 | 0.063 | 0.073 | 0.052 | OOM | OOM | OOM | **0.226** | 0.413 | 0.569 | 0.489 | 5.56 |
| Kron | **0.069** | **0.028** | 0.056 | 0.378 | 0.060 | 0.064 | OOM | OOM | OOM | 0.246 | 0.413 | 0.554 | 0.491 | 6.12 |
| UGC(fea.) | 0.224 | 0.340 | 0.208 | 0.179 | 0.145 | **0.016** | **0.014** | EOOM | EOOM | 13.8 | 7.594 | 0.420 | 0.534 | 9.83 |
| UGC(fea+Ad) | 0.130 | 0.070 | 0.050 | **0.004** | **0.004** | 0.018 | 0.0153 | EOOM | EOOM | 0.546 | **0.409** | **0.215** | **0.204** | **0.075** |

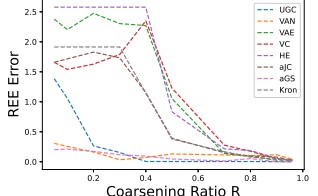
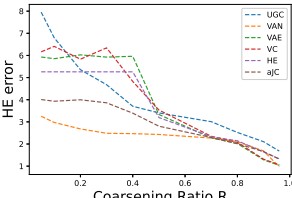
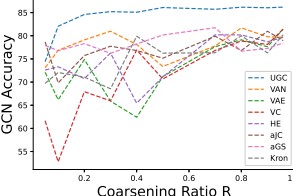

Figure 5: This figure compares graph coarsening methods in terms of REE, HE, and GCN accuracy on the Pubmed dataset.

graph, hence the term OOM. Figure 5 illustrates the trends for eigen error, hyperbolic error and GCN accuracy for different methods as the coarsening ratio is altered.

**LSH Similarity and $\epsilon$-Bounded Results** The LSH family used in our framework is based on p-stable distributions $\mathcal{D}$ see Appendix A. This ensures that the probability of two nodes going to the same super-node is directly related to the distance between their features (augmented features $F$ for UGC).

**Theorem 4.1** *As given in [43], the probability that two nodes $v$ and $u$ will collide and go to a super-node under a hash function drawn uniformly at random from a 2-stable distribution is inversely proportional to $c = ||v - u||_2$ and it is represented by $p(c) = Pr_{w,b}[h_{w,b}(v) = h_{w,b}(u)] = \int_0^r \frac{1}{c} f_p\left(\frac{t}{c}\right)\left(1 - \frac{t}{r}\right) dt$.*

In our experiments, we empirically validated the Theorem 4.1. We examined if the feature distance between any node pair was below a specific threshold, and then using the coarsening matrix $\mathcal{C}$ given by UGC, we verified if they shared the same super-node or not. Our evaluation involved counting successful matches, where nodes belonged to the same super-node, and failures, where they did not. We subsequently calculated a probability measure based on these counts. Figure 6a and 6b plot this probabilistic function for two datasets, namely *Cora* and *Citeseer* as a function of distance between two nodes. Re-visiting the Definition 2.1 for the *Cora* dataset, we denote our LSH family as $\mathcal{H}(1, 3, 1, 0.20)$. Suppose $d$ denotes the distance between the nodes $\{u, v\}$. In the notation $\mathcal{H}(1, 3, 1, 0.20)$, this implies that if $d \leq 1$, there is a 100% probability that $u, v$ will be grouped into the same super-node. Conversely, if $d > 3$, the probability of $\{u, v\}$ being grouped into the same super-node is 20%. Figure 6c plots different values of $\epsilon$ at different coarsening ratios. We used Equation 6 for updating the augmented feature matrix $F$ given by UGC and as mentioned, we got $\epsilon \leq 1$ similarity guarantees for the coarsened graph. Hence proving Theorem 3.2.

**Scalable Training of Graph Neural Networks.** Graph neural networks (GNNs), tailored for non-Euclidean data [44–46], have shown promise in various applications [47, 48]. However, scalability remains a challenge. Building on [34], we investigate how our graph coarsening approach can enhance GNN scalability for training, bridging the gap between GNNs and efficient processing of large-scale data.

*GNN parameter details.* We employed a single hidden layer GCN model with standard hyperparameters values [13] see Appendix H for the node-classification task. Coarsened graph $\mathcal{G}_c$ is used to train the GCN model, and all the predictions are made on test data from the original graph. The relation between coarsening ratio and accuracy is evident from Table 9 in Appendix H. Specifically, as we progressively coarsen the graph, a slight decrease in accuracy values becomes noticeable. Hence, there will always be a trade-off when it comes to the

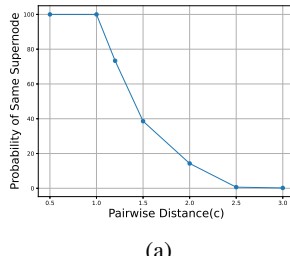 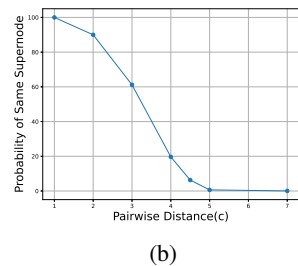 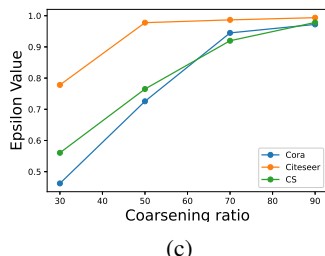

| (a) | (b) | (c) |

Figure 6: a) Cora and b) Citeseer demonstrate the inverse relationship between the probability of two nodes belonging to the same super-node and the distance between them. c) plots the $\epsilon$ values ($\leq 1$) for Cora, Citeseer, and CS datasets.

Table 4: This table illustrates the accuracy of GCN model when trained with 50% coarsen graph. UGC demonstrated superior performance compared to existing methods in 7 out of the 9 datasets.

| Data/Method | Cora | DBLP | PubMed | Physics | Squirrel | Cham. | Cor. | Texas | Film |
|---|---|---|---|---|---|---|---|---|---|
| Var.Neigh. | 79.75 | 77.05 | 77.87 | 93.74 | 19.67 | 20.03 | 52.49 | 34.51 | 15.67 |
| Var.Edges | 81.57 | **79.93** | 78.34 | 93.86 | 20.22 | 29.95 | 55.32 | 30.59 | 21.8 |
| Var.Clique | 80.92 | 79.15 | 73.32 | 92.94 | 19.54 | 31.92 | 58.8 | 33.92 | 20.35 |
| Heavy Edge | 79.90 | 77.46 | 74.66 | 93.03 | 20.36 | 33.3 | 54.67 | 29.18 | 19.16 |
| Alg. Dis. | 79.83 | 74.51 | 74.59 | 93.94 | 19.96 | 28.81 | **59.91** | 18.61 | 19.23 |
| Aff. GS | 80.20 | 78.15 | 80.53 | 93.06 | 20.00 | 27.58 | 54.06 | 21.18 | 20.34 |
| Kron | 80.71 | 77.79 | 74.89 | 92.26 | 18.03 | 29.1 | 55.02 | 31.14 | 17.41 |
| UGC(fea.) | 83.92 | 75.50 | **85.65** | 94.70 | 20.71 | 29.9 | 55.6 | 52.4 | 22.6 |
| UGC(fea+Ad) | **86.30** | 75.50 | 84.77 | **96.12** | **31.62** | **48.7** | 54.7 | **57.1** | **25.4** |

coarsening ratio and quality of the reduced graph. To emphasize the contribution of UGC in terms of both computational time and node-classification accuracy, we have included Figure 7.

This figure illustrates the improvements in computational time and the corresponding changes in accuracy values when compared to the currently best-performing model across various datasets. Table 4 compares the accuracy among all the approaches with all datasets when they are coarsened down by 50%. UGC demonstrated superior performance compared to existing methods in 7 out of the 9 datasets. We have used t-SNE [49] algorithm for visualization of predicted node labels shown in Figure 10 in Appendix H. It is evident that even with highly coarsened graph training, the GCN model can maintain its accuracy.

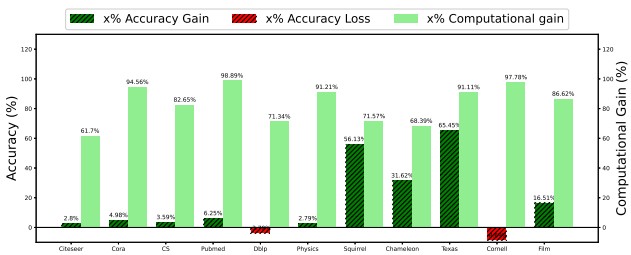

Figure 7: Computational and accuracy gains of UGC. In the bar plot, dashed bars represent the gain or loss in accuracy when compared to the existing best-performing method, while plain bars indicate the computational gains. All datasets are coarsened down by 50%.

**UGC is Model-Agnostic.** While our initial validation utilized GCN to assess the quality of our coarsened graph $\mathcal{G}_c$ our framework is not bound to any specific GNN architecture. We extended our evaluations to include other prominent graph neural network models. Results from three diverse models, namely GCN [13], GraphSage [40], GIN [50], and GAT [51], have been incorporated into our analysis. All the models were trained using 50% coarsened graph $\mathcal{G}_c$. Results from Table 3 demonstrate the robustness and model-agnostic nature of UGC. Refer to Table 7 in

Table 3: This table demonstrates UGC's model-agnostic nature, as it doesn't rely on any specific GNN model.

| Model/Data | Cora | Pubmed | Physics | Squirrel |
|---|---|---|---|---|
| GCN | 86.30 | 84.77 | 96.12 | 31.62 |
| GraphSage | 69.39 | 85.72 | 94.49 | 61.23 |
| GIN | 67.23 | 84.12 | 85.15 | 44.72 |
| GAT | 74.21 | 84.37 | 92.60 | 48.75 |

Appendix H for a comprehensive analysis of node classification accuracy results for various GNN models. We believe this flexibility further enhances the applicability and utility of our proposed framework in various graph-based applications.

**Gained Performance on Heterophilic Graphs.** Existing work for GC is focused on homophilic datasets. A notable contribution of our framework is its ability to generalize to all datasets, including heterophilic datasets as well. Building upon the observations made in Table 2 and Table 4 our methods, UGC (feat) and UGC (aug. feat.), showcase notable improvements in both node classification accuracy and REE values when applied to heterophilic datasets. A comparison of these results reveals that conventional approaches demonstrate poor node-classification accuracy on heterophilic graphs. In contrast, our UGC (features) method achieves substantial accuracy enhancements, surpassing the performance of these traditional approaches. Furthermore, the true potential of our approach becomes evident with augmented features $F$ i.e., UGC (aug. feat.). This approach exhibits remarkable accuracy gains, outperforming all other methods by a considerable margin, signifying the importance of augmented features $F$.

## 5   Conclusion

In this paper, we present a framework **UGC** for reducing a larger graph to a smaller graph. We use hashing of augmented node features inspired by Locality Sensitive Hashing (LSH). As expected, the benefits of LSH are also reflected in the proposed coarsening algorithm. To the best of our knowledge, it is the fastest algorithm for graph coarsening. Through extensive experiments, we have also shown that our algorithm is not only fast but also preserves the properties of the original graph. Furthermore, it is worth noting that UGC represents the first work in the domain of graph coarsening for heterophilic datasets. This framework addresses the unique challenges posed by heterophilic graphs and has demonstrated a significant increase in node classification accuracy following graph coarsening. In conclusion, we believe that our framework is a major contribution to the field of graph coarsening and offers a fast and effective solution for simplifying large networks. Our future research goals include the exploration of different hash functions and novel applications for the framework.

## 6   Acknowledgement

Mohit Kataria acknowledges the generous grant received from Google Research India to sponsor his travel to NeurIPS 2024. Additionally, this work is supported by DST INSPIRE faculty grant MI02322G and Yardi-ScAI, IIT Delhi research fund.

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

# Appendix

## A  Stable Distribution

A distribution $\mathcal{D}$ over $\mathcal{R}$ is called p-stable if there exists p $\geq$ 0 such that for any n real numbers $v_1....v_n$ and i.i.d. variables $X_1....X_n$ with distribution $\mathcal{D}$, the random variable $\sum_i v_i X_i$ has the same distribution as the variable $(\sum_i |v_i|^p)^{1/p} X$ where $X$ is a random variable with distribution $\mathcal{D}$ [52]. It is known [53] that stable distributions exist for p $\in$ (0,2].

- *Cauchy distribution* $\mathcal{D}_c$, defined by the density function $c(x) = \frac{1}{\pi} \frac{1}{1+x^2}$, is 1-stable.
- *Gaussian (normal) distribution* $\mathcal{D}_g$, defined by the density function $g(x) = \frac{1}{\sqrt{2\pi}} e^{\frac{-x^2}{2}}$ is 2-stable.

However, it is known [54] that one can create p-stable random variables effectively from two independent variables distributed uniformly across [0,1] despite the lack of closed-form density and distribution functions.

Stable distributions have diverse applications across various fields (see survey [55] for details). In computer science, they are utilized for "sketching" high-dimensional vectors, as demonstrated by Indyk ([43]). The key property of p-stable distributions, mentioned in the definition, enables a sketching technique for high-dimensional vectors. This technique involves generating a random vector **w** of dimension **d**, with each entry independently chosen from a p-stable distribution. Given a vector **v** of dimension d, the dot product $w \cdot v$ is also a random variable. A small collection of such dot products, corresponding to different w's, is termed as the sketch of the vector v and can be used to estimate $||v||_p$ [43]. However, instead of using the sketch to estimate the vector norm, we are using it to assign hash values to each vector. These values map each vector to a point on the real line, which is then split into equal-width segments to represent buckets. If two vectors v and If you are close, they will have a small difference between $l_p$ norms $\|v - u\|_p$, and they should collide with a high probability

## B  Algorithm for UGC-FL

---
**Algorithm 2** UGC-FL: Universal Graph Coarsening with feature re-learning

---
**Require:** Input $\mathcal{G}(V, A, X)$, $l \leftarrow$ Number of Projectors, $r \leftarrow binWidth$
1: $\mathcal{G}_c(\widetilde{V}, \widetilde{A}, \widetilde{F}), \mathcal{C} = UGC(\mathcal{G}, l, r)$
2: $\alpha = \frac{|\{(v,u) \in E : y_v = y_u\}|}{|E|}$; $\alpha$ is heterophily factor, $y_i \in \mathbb{R}^N$ denotes node labels, $E$ denotes edge list
3: $F = \{(1 - \alpha) \cdot X \oplus \alpha \cdot A\}$ Augmented features
4: $\widehat{F} = \left(\frac{2}{\alpha} \mathcal{C}^T L \mathcal{C} + \mathcal{C}^T \mathcal{C}\right)^{-1} \mathcal{C}^T F$
5: return $\mathcal{G}_c(\widetilde{V}, \widetilde{A}, \widehat{F}), \mathcal{C}$

---

## C  Size of coarsened graph Controlled by Bin-Width

This section discusses the impact of bin-width on the coarsening ratio see Figure 8. Algorithm 3 outlines the procedure for determining the appropriate bin-width value that corresponds to a desired coarsening ratio. The parameter bin-width $r$ decides the size of the coarsened graph $\mathcal{G}_c$. For a particular coarsening ratio $R$, we find the corresponding $r$ by divide and conquer approach on the real axis, which is similar to binary search. Algorithm 3 shows the method by which we find the $r$ for any given $R$ for $\mathcal{G}_c$. Figure 8 shows the relation of $r$ with $R$ for two datasets: a) Cora, and Coauthor CS. It is observed that the $R$ increases as the $r$ increases. For each dataset, $r$ is a hyper-parameter that needs to be calculated only once, and hence its computational cost is not included in the reported time complexity.

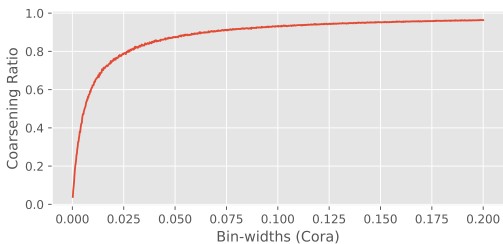 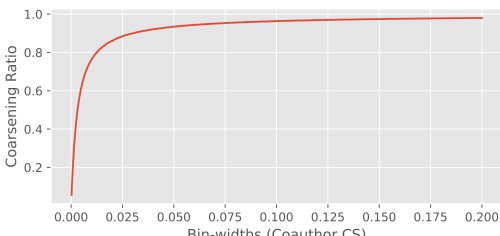

Figure 8: This figure shows the trend of coarsening ratio as the bin-width increases on two datasets: Cora and Coauthor CS.

---

**Algorithm 3** Bin-width Finder

---

**Require:** Input $\mathcal{G}(V, A, X)$, $L \leftarrow$ Number of Projectors, $R \leftarrow$ Coarsening Ratio, $p \leftarrow$ precision of coarsening
**Ensure:** bin-width $h$
1: $r \leftarrow 1, ratio \leftarrow 1, N \leftarrow \|V\|$
2: **while** $|R - ratio| > p$ **do**
3:    **if** $ratio > R$ **then**
4:       $r \leftarrow r * 0.5$
5:    **else**
6:       $r \leftarrow r * 1.5$
7:    $n \leftarrow \text{UGC}(\mathcal{G}, L, r)$; $n$ denotes number of supernodes in $\mathcal{G}_c$
8:    $ratio \leftarrow (1 - \frac{n}{N})$
9: **return** $r$

---

## D  Proof of Theorem 4.1

Let $f_p(t)$ denote the probability density function of the absolute value of our stable distribution(Normal distribution), and let $c = ||v - u||_p$ for two node vectors v, u, and r is the bin-width. Since we have a random vector w from our stable distribution, $v.w - u.w$ is distributed as $cX$ where X is a random variable from our stable distribution. Therefore our probability function is

$$p(c) = Pr_{a,b}\left[h_{a,b}(v) = h_{a,b}(u)\right] = \int_0^r \frac{1}{c} f_p\left(\frac{t}{c}\right)\left(1 - \frac{t}{r}\right) dt \tag{8}$$

For a fixed bin-width $r$ the probability of collision decreases monotonically with $c = ||v - u||_2$. For Definition, 2.1 the hash family will be $(r_1, r_2, p_1, p_2)$-sensitive where $p_1 = p(1)$ and $p_2 = p(c)$ for $\frac{r_2}{r_1} = c$.

For 2-stable distribution $f_p(x) = \frac{2}{\sqrt{2\pi}} e^{-x^2/2}$. Equation 9 will be

$$p(c) = \frac{2}{\sqrt{2\pi}} \int_0^r \frac{1}{c} e^{-\left(\frac{1}{c}\right)^2/2} dt - \frac{2}{\sqrt{2\pi}} \int_0^r \frac{1}{c} e^{-\left(\frac{1}{c}\right)^2/2} \frac{t}{r} dt \tag{9}$$

$$= S_1(c) - S_2(c)$$

Note that $S_1(c) \le 1$.

$$S_2(c) = \frac{2}{\sqrt{2\pi}} \cdot \frac{c}{r} \int_0^r e^{-\left(\frac{t}{c}\right)^2/2} \frac{t}{c^2} dt \tag{10}$$

$$S_2(c) = \frac{2}{\sqrt{2\pi}} \cdot \frac{c}{r} \int_0^{\frac{\mathbb{R}^2}{(2c^2)}} e^{-y} dy \tag{11}$$

$$S_2(c) = \frac{2}{\sqrt{2\pi}} \cdot \frac{c}{r}\left[1 - e^{-\frac{\mathbb{R}^2}{(2c^2)}}\right] \tag{12}$$

We have $p(1) = S_1(1) - S_2(1) \geq 1 - e^{\frac{\mathbb{R}^2}{2}} - \frac{2}{\sqrt{2\pi}r} \geq 1 - \frac{A}{r}$, for some constant A > 0. This implies that the probability that u collides with v is at least $(1 - \frac{A}{r}) \approx e^{-A}$. Thus the algorithm is correct with the constant probability.

If $c^2 \leq \frac{\mathbb{R}^2}{2}$, then we have

$$p(c) \leq 1 - \frac{2}{\sqrt{2\pi}} \frac{c}{r} (1 - \frac{1}{e}) \tag{13}$$

## E  Additional experiments for LSH scheme

We have further validated our theoretical results through a secondary experiment. This LSH family which we discussed above says as the distance between two nodes increases, the likelihood of them being assigned to the same bin decreases, hence we will have more number of super-nodes now. By increasing the bin-width, we can effectively reduce the number of super-nodes. This phenomenon is evident when considering the average distance between node pairs in various graphs and the corresponding bin-width required to achieve a 30% coarsening ratio. The table below illustrates these findings:

Table 5: Average Distance and Bin-Width for 30% Coarsening

| Dataset | Average Distance | Bin-Width |
|---------|------------------|-----------|
| Citeseer | 7.748 | 0.0029 |
| Cora | 5.810 | 0.0021 |
| Dblp | 3.168 | 0.000068 |
| Pubmed | 0.540 | 0.000025 |

The results in the table clearly demonstrate that as the average distance between nodes increases, the required bin-width also increases when maintaining the same coarsening ratio. This observation highlights the importance of considering the distance metric and bin-width selection during the graph coarsening process to effectively control the number of super-nodes and achieve desired coarsening ratios. Figure 8 shows the trend of coarsening ratio when we change bin-width.

## F  System Specifications:

All the experiments conducted for this work were performed on an Intel Xeon W-295 CPU and 64GB of RAM desktop using the Python environment.

## G  Datasets

Table 6: Summary of the datasets. H.R shows heterophily factor.

| Data | Nodes | Edges | Features | Class | H.R($\alpha$) |
|------|-------|-------|----------|-------|---------------|
| Cora | 2,708 | 5,429 | 1,433 | 7 | 0.19 |
| Citeseer | 3,327 | 9,104 | 3,703 | 6 | 0.26 |
| DBLP | 17,716 | 52,867 | 1,639 | 4 | 0.18 |
| CS | 18,333 | 163,788 | 6,805 | 15 | 0.20 |
| PubMed | 19,717 | 44,338 | 500 | 3 | 0.20 |
| Phy. | 34,493 | 247,962 | 8,415 | 5 | 0.07 |
| Flickr | 89,250 | 899,756 | 500 | 7 | 0.69 |
| Reddit | 232,965 | 114.61M | 602 | 41 | 0.25 |
| Yelp | 716,847 | 13.95M | 300 | 100 | |
| Texas | 183 | 309 | 1703 | 5 | 0.91 |
| Cornell | 183 | 295 | 1703 | 5 | 0.70 |
| Film | 7600 | 33544 | 931 | 5 | 0.78 |
| Squirrel | 5201 | 217073 | 2089 | 5 | 0.78 |
| Chameleon | 2277 | 36101 | 2325 | 5 | 0.75 |

# H Application of coarsened graph for GNNs

This section contains additional results related to the scalable GNN training. Figure 9 shows the GNN training pipeline. Figure 10 shows the visualization of GCN predicted nodes when training is done using the coarsened graph.

We used four GNN models, namely GCN, GraphSage, GIN, and GAT. Table 7 contains node classification accuracy results for across various methodologies employing different GNN models. Table 8 contains parameter details used in UGC across different GNN models. We have used these parameters across all methods.

Table 7: Evaluation of node classification accuracy of different GNN models when trained with 50% coarsen graph.

| Dataset | Model | Var.Neigh | Var.Edges | Var.Clique | Heavy Edge | Alg. Dis. | Aff. GS | Kron | UGC |
|---|---|---|---|---|---|---|---|---|---|
| Chameleon | GCN | 20.03 | 29.95 | 31.92 | 33.30 | 28.81 | 27.58 | 29.10 | **48.70** |
| | GraphSage | 20.03 | 20.02 | 22.05 | 23.03 | 19.88 | 20.02 | 27.62 | **58.86** |
| | GIN | 20.22 | 19.53 | 25.25 | 19.98 | 18.20 | 18.06 | 21.50 | **54.92** |
| | GAT | 22.94 | 19.33 | 26.44 | 21.95 | 23.72 | 18.06 | 21.95 | **55.58** |
| Squirrel | GCN | 19.67 | 20.22 | 19.54 | 20.36 | 19.96 | 20.00 | 18.03 | **31.62** |
| | GraphSage | 19.87 | 20.00 | 20.03 | 20.03 | 19.93 | 20.00 | 19.98 | **57.60** |
| | GIN | 18.54 | 19.65 | 18.98 | 21.65 | 19.47 | 18.29 | 20.56 | **35.64** |
| | GAT | 20.90 | 18.56 | 20.68 | 19.93 | 20.46 | 20.05 | 20.08 | **32.28** |
| Film | GCN | 15.67 | 21.80 | 20.35 | 19.16 | 19.23 | 20.34 | 17.41 | **25.40** |
| | GraphSage | 22.32 | 26.05 | 24.01 | 21.49 | 21.88 | 21.50 | **23.73** | 21.12 |
| | GIN | **24.20** | 23.51 | 17.51 | 11.49 | 13.90 | 21.93 | 18.04 | 21.12 |
| | GAT | 17.50 | 21.73 | 17.82 | 21.18 | 17.94 | 17.40 | **24.15** | 21.71 |
| Pubmed | GCN | 77.87 | 78.34 | 73.32 | 74.66 | 74.59 | 80.53 | 74.89 | **84.77** |
| | GraphSage | 78.85 | 62.73 | 67.18 | 60.11 | 63.09 | 71.25 | 62.00 | **83.76** |
| | GIN | 74.77 | 39.29 | 46.19 | 35.97 | 32.13 | 49.63 | 39.29 | **76.36** |
| | GAT | 75.22 | 72.63 | 74.81 | 60.04 | 69.47 | 59.76 | 71.92 | **83.56** |
| Physics | GCN | 93.74 | 93.86 | 92.94 | 93.03 | 93.94 | 93.06 | 92.26 | **96.12** |
| | GraphSage | OOM | OOM | OOM | OOM | OOM | OOM | OOM | OOM |
| | GIN | OOM | OOM | OOM | OOM | OOM | OOM | OOM | OOM |
| | GAT | 92.04 | 91.80 | 91.48 | 91.80 | 92.94 | 93.33 | 91.60 | **93.80** |
| DBLP | GCN | 77.05 | **79.93** | 79.15 | 77.46 | 74.51 | 78.15 | 77.79 | 75.50 |
| | GraphSage | 68.54 | 60.17 | **74.17** | 72.70 | 72.19 | 71.81 | 71.76 | 68.25 |
| | GIN | 35.84 | 33.93 | 35.12 | 24.16 | 51.47 | 47.30 | 42.24 | **55.28** |
| | GAT | 70.20 | **74.07** | 72.82 | 71.35 | 71.17 | 76.12 | 72.27 | 73.49 |
| Cora | GCN | 79.75 | 81.57 | 80.92 | 79.90 | 79.83 | 80.20 | 80.71 | **86.30** |
| | GraphSage | 70.49 | 68.48 | 70.16 | 69.17 | 72.26 | 67.77 | **73.20** | 69.39 |
| | GIN | 47.65 | 35.03 | 52.91 | 34.00 | 63.05 | 23.49 | 48.56 | **67.23** |
| | GAT | 69.26 | 74.02 | **75.92** | 68.95 | 73.09 | 73.83 | 73.24 | 74.21 |

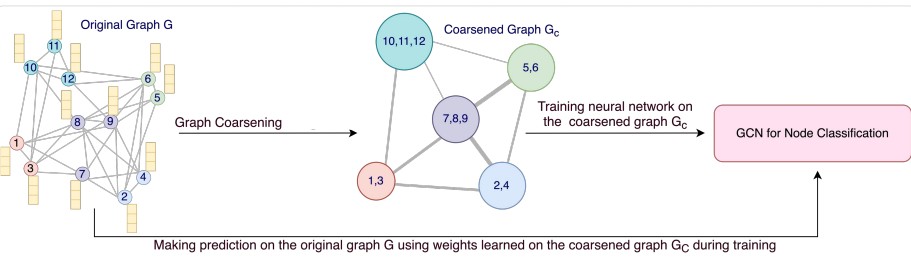

Figure 9: GCN training pipeline

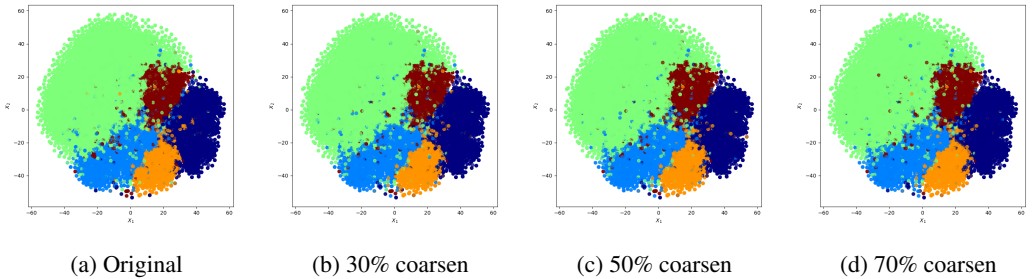

| (a) Original | (b) 30% coarsen | (c) 50% coarsen | (d) 70% coarsen |

Figure 10: Visualization of GCN predicted nodes when training is done using the coarsened graph of Physics dataset.

Table 8: GNN model parameters.

| MODEL | HIDDEN LAYERS | LEARNING RATE | DECAY | EPOCH |
|---|---|---|---|---|
| GCN | $\{64, 64\}$ | 0.003 | 0.0005 | 500 |
| GRAPHSAGE | $\{64, 64\}$ | 0.003 | 0.0005 | 500 |
| GIN | $\{64, 64\}$ | 0.003 | 0.0005 | 500 |
| GAT | $\{64, 64\}$ | 0.003 | 0.0005 | 500 |

Table 9: We report the accuracy of GCN on node classification after coarsening by UGC at different ratios.

| Ratio/Data | Cora | DBLP | Pub. | Phy. |
|---|---|---|---|---|
| 30 | 86.30 | 75.50 | 85.65 | 96.70 |
| 50 | 86.30 | 75.50 | 84.77 | 96.12 |
| 70 | 84.63 | 74.82 | 80.57 | 92.43 |

We randomly split data in 60%, 20%, 20% for the training-validation-test.

# I  Proof of Theorem 3.1

**Theorem I.1** *Given a Graph $G$ and a coarsened graph $G_c$ they are said to be $\epsilon$ similar if there exists some $\epsilon \geq 0$ such that:*

$$(1 - \epsilon)\|x\|_L \leq \|x\|_{L_c} \leq (1 + \epsilon)\|x\|_L \tag{14}$$

*where $L$ and $L_c$ are the Laplacian matrices of $G$ and $G_c$ respectively.*

**Proof:** Let S be defined such that $L = S^T S$, by Cholesky's decomposition:

$$|\|x\|_L - \|x_c\|_{L_c}| = |\|Sx\|_2 - \|SP^+ Px\|_2| \tag{15}$$

$$\leq \|Sx - SP^+ Px\|_2 = \|x - \widetilde{x}\|_L \leq \|x\|_L \tag{16}$$

The conversion from $L^{th}$−norm to $2^{nd}$−norm or vice-versa is as follows:

$$\|x\|_L = \sqrt{x^T L x} = \sqrt{x^T S^T S x} = \|S\|_2$$

# J  Proof of Bounded Theorem 3.2

**Theorem J.1** *Given a graph $\mathcal{G}(L, F)$, a coarsened graph $\mathcal{G}_c(L_c, F_c)$, the enhanced features $\widetilde{F}$ obtained by enforcing smoothness condition. The original graph $\mathcal{G}(L, F)$ and a coarsened graph $\mathcal{G}_c(L_c, \widetilde{F})$, are said to be $\epsilon$ similar with $0 < \epsilon \leq 1$*

$$(1 - \epsilon)\|F\|_L \leq \|\widetilde{F}\|_{L_c} \leq (1 + \epsilon)\|F\|_L \tag{17}$$

where $L$ and $L_c$ are the Laplacian matrices, $F$ and $F_c$ are the augmented feature vector given by UGC, $\widetilde{F}$ is the relearned enhanced features by enforcing the smoothness condition discussed in Equation 6.

**Proof:**

$$\left| \|F\|_L - \|\widetilde{F}\|_{L_c} \right| = \left| \sqrt{tr(F^T L F)} - \sqrt{tr(\widetilde{F}^T L_c \widetilde{F})} \right| \tag{18}$$

As $L$ is a positive semi-definite matrix we can write $L = S^T S$ using Cholesky's decomposition and by writing $L_c = C^T L C$ we get,

$$= \left| \sqrt{tr(F^T S^T S F)} - \sqrt{tr(\widetilde{F}^T C^T S^T S C \widetilde{F})} \right| \tag{19}$$

$$= \left| \|SF\|_F - \|SP^\dagger PF\|_F \right| \tag{20}$$

$$\leq \|SF - SP^\dagger PF\|_F \tag{21}$$

$$\leq \epsilon \|F\|_L \tag{22}$$

Using the new update rule of $\|\widetilde{F}\|_{L_c}$ we have $\widetilde{F}_{L_c} \leq \|F\|_L$, we get

$$\epsilon = \frac{\left| \|F\|_L - \|\widetilde{F}\|_{L_c} \right|}{\|F\|_L} \leq 1 \tag{23}$$

where $\epsilon \leq 1$ refer [14] for more details. See Figure 6 which plots different values of $\epsilon$ at different coarsening ratios. As mentioned for fixed values of $\alpha$ we got $\epsilon \leq 1$ similarity guarantees for the coarsened graph.

## K   Importance of Augmented Features

See Figure 12 which showcases the importance of considering the augmented feature vector. It can be seen from the figure that when coarsened using Augmented features super-nodes have more intra-node similarity.

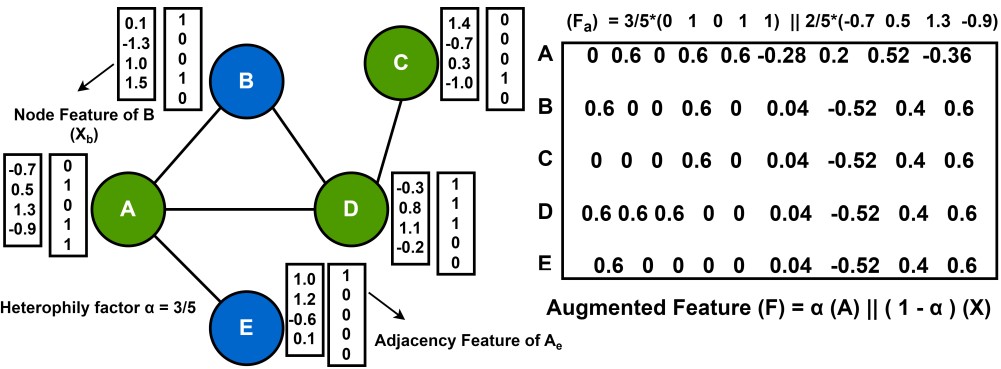

Figure 11: A toy example illustrating the computation of augmented features of a given graph.

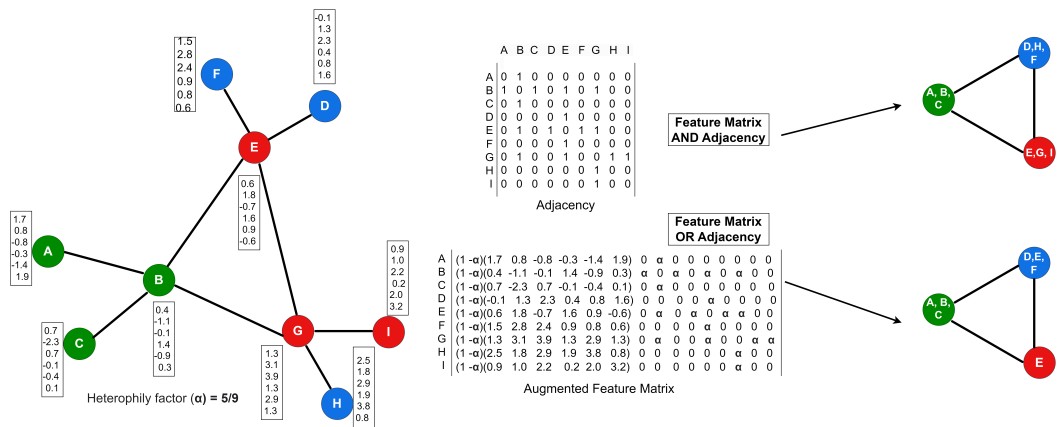

Figure 12: This figure highlights the significance of the augmented vector and showcases coarsening outcomes, specifically when coarsening is performed solely using the adjacency or feature matrix compared to when the augmented matrix is taken into account.

