# OpenReview forum: "UGC: Universal Graph Coarsening"
_NeurIPS.cc/2024/Conference — NeurIPS 2024 poster_

### Official Review · Reviewer_QBq8 · 2024-06-18

**Soundness:** 2
**Presentation:** 1
**Contribution:** 2
**Rating:** 5
**Confidence:** 4

**Summary:**

Graph coarsening aims at scaling an original large graph into a small graph. This paper proposes a graph coarsening method which was designed to be equally suitable for homophilic and heterophilic datasets, specifically, aggregating node clusters identified by a hash function. This paper is one of the pioneering works exploring graph coarsening in heterophilic graph datasets, and the proposed solution seems promising. However, the current formulation of the proposed method and some of the claims in the paper are not rigorous and solid enough.

**Strengths:**

-	This paper is one of the pioneering works exploring graph coarsening in heterophilic graph datasets.
-	Aggregating hypernodes through a hash function instead of neighbors is a promising solution to perform heterophilic graph coarsening.

**Weaknesses:**

-	The presentation of this paper is unclear. Check out `Questions’.
-	The current formulation of the proposed `Hash functions’ is ambiguous and biased. Check out `Questions’.
-	Some claims were in fact wrong. Check out `Questions’.

**Questions:**

-	Line 4. `vital insights’, `essential features’ in line 5. and ‘vital information’ in line 20. are too vague. I suggest being more specific and keeping the phrasing unchanged.
-	Line 67-68. `The Graph Coarsening(GC) problem \bf{may be} thought of as learning a coarsening matrix….’ Is it or is it not? A formal definition of the Graph Coarsening problem is required whether cited from others or by you.
-	Line 128-129. `Due to this, these methods are extremely computationally demanding and defeat the purpose of reducing the original graph.’ This statement suggests that ‘GNN-based Graph Condensation’ is useless. I suggest being more objective.
-	Line 143-144.` The concepts of homophily and heterophily in graphs are primarily concerned with whether the edges between two nodes align with their respective features.’ is not correct. Recall that homophily and heterophily are defined by whether edges align with labels, not features. You need to state a hypothesis or experimentally verify that: `the same labels imply similar features’.
-	Line 155-160. The current formulation hash function is biased to node degree distribution since you concat the adj with node feature. Consider this example that a graph has 1 node whose degree is 999 and other 999 nodes’ degree is 1 (a radial graph). In this case, the hash value of the degree 999 node will be significantly larger than the other nodes, e.g. 90-99 to 1-10, and setting `r’ parameter only makes it worse. Meanwhile, the 11-89’th hypernode is empty. This is an extreme case of long-tail distribution of node degrees, however, simple concatenation inevitably causes the number of aggregated nodes in the hypernode to be uneven, specifically strongly correlated with the node distribution.
-	Again, concat the adj with node feature is ambiguous. Considering that the hash result will change w.r.t. the change of index order of nodes.
-	What is ‘|||’ in the proof of Theorem 3.1 in line 575? Moreover, the derivation in the proof is speculative. You need to explain clearly why each step of the inequality holds.
-	Line 180-181. `It means that the adjacency matrix A~ has a substantially smaller number of non-zero elements than A.’ is not correct. Consider a chain graph with N nodes, N-1 edges, and if the head node and the tail node were aggregated, it became a circle with N-1 nodes and N-1 edges, where the number of edges remains unchanged.
-	Where is the introduction of baselines from `Var.Neigh.’ to ` Kron’? If somewhere else, I suggest moving to Chap experiments.
-	Figure 7 is confusing. Present `actual time saved’ (time by downstream task on original graph – time by downstream task on coarsened graph + time by coarsening the graph) together with downstream task performance loss/gain by your method and baselines will be more meaningful.

**Limitations:**

The authors have adequately addressed the limitations.

---

> ### Author Rebuttal · Authors · 2024-08-07
>
> We thank the reviewer for their valuable comments and insights and for taking the time to go through our paper.
>
> **Ques 1)** Regarding .. *Line 4. vital insights’ *
>
> **Ans 1)**  We thank the reviewer for the suggestion.  By *vital insights* and *vital information*, we mean retaining the basic statistics of the graphs, such as spectral properties and $\epsilon$-similarity in the coarsened graph. This ensures that downstream processes are more efficient and effective. Given the opportunity, we can integrate these clarifications into the manuscript.
>
> **Ques 2)** Regarding *Line 67-68. The Graph Coarsening(GC) problem ..*
>
> **Ans 2)** The Graph Coarsening (GC) problem is indeed about learning a coarsening matrix. This definition is well-established in the literature [1, Section 2], [2, Section 2.2]. The objective and problem formulation are also discussed in Section 2.
>
> [1] Hashemi, Mohammad, et al. "A comprehensive survey on graph reduction: Sparsification, coarsening, and condensation." arXiv preprint arXiv:2402.03358 (2024).
>
> [2] Kumar, Manoj, et al. "Featured graph coarsening with similarity guarantees." International Conference on Machine Learning. PMLR, 2023.
>
> **Ques 3)** Regarding *Line 128-129. Due to this, these methods ....*
>
> **Ans 3)** We thank the reviewer for the suggestion. The main reason behind this statement was the following results, where we compared the coarsening time using GCond with the whole graph training time. As it can be seen from the table, the time it takes to coarsen the graph is multiple times higher than the training time. However, the above-mentioned statement can be rephrased as :
>
> *...these methods are extremely computationally demanding and may not be suitable for the scalability of GNN models. However, these methods can be beneficial for other tasks, like solving storage and visualization issues.*
>
> GCond accuracy and time
> Data | GCond Acc | GCond Coarsen Time |GCN training time on original graph|UGC Coarsening Time(x Fast compared to GCond) |
> |--|--|--|--|--|
> |Cora|80.43|2640|25.77|0.41(x6440)|
> |Pubmed|76.98|1620|114.55|1.62(x1000)|
> |Physics|OOM|-|1195.56|6.4|
> |DBLP|82.63|25500|174.10|1.86(x13710)|
> |Squirrel|59.64|7860|228.52|2.14(x3673)|
> |Chameleon|52.29|7740|54.34|0.49(x15796)|
>
> **Ques 4)** Regarding *Line 143-144.The concepts of homophily and heterophily .....*
>
> **Ans 4)** We understand the statement mentioned can be misleading and therefore we have corrected the line to the following:
>
>  *The concepts of homophily and heterophily in graphs are primarily concerned with whether the edges between two nodes align with their respective labels.*
>
> **Ques 5)** Regarding.. *Line 155-160. The current formulation hash function ... Consider this example that a graph has 1 node whose degree is 999 and......*
>
> **Ans 5)** We appreciate your concern. To clarify, in the example cited by the reviewer, the node with a degree of 999 will indeed produce a higher hash value and will be collapsed into a super-node. This is reasonable because it is an important node and should be assigned to a different super-node. It is important to note that we ensure no super-node is empty during the formulation of the super-nodes. If there is no hash value between the range of 11-89, we will not create a super-node for this range. For the other nodes with a degree of 1, as these nodes are connected to the same high-degree node, the distinction between the hash values will be governed by their node feature vectors. The super-node index is determined by setting the appropriate bin-width value "r".
>
> **Ques 6)** Regarding.. *Again, concat the adj with node feature is ambiguous...*
>
> **Ans 6)** We appreciate your concern. To clarify, the order of the nodes is fixed and consistent across all *l* projectors at the start of the coarsening and does not change during the process. Once the node order is established, the LSH framework's locality-preserving property ensures that the hashing process remains stable.
>
> **Ques 7)** Regarding. *What is ‘|||’ in the proof of Theorem 3.1 in l......*
>
> **Ans 7)** We thank the reviewer for pointing out the typo. $||.||_p$ denotes $p^{th}$ norm. We have corrected the Proof of Theorem 3.1 here is the revised version:
>
> **Proof:** Let S be defined such that L = $S^TS$, by  Cholesky’s decomposition:
>
>
> $| \lVert x\rVert_{L} - \lVert x_c\rVert_{L_c} | = | \lVert Sx\rVert_{2} - \lVert SP^+Px\rVert_{2}| $
>
> From Modulus inequality property,
>
> \begin{gather}
>     \leq ||Sx - SP^+Px||_2 = ||x - \widetilde{x}||_L \leq ||x||_L
> \end{gather}
>
> The conversion from $L-norm$ to $2^{nd}-norm$ or vice-versa is as follows:
>
>  $\lVert x\lVert_L = \sqrt{x^T L x} = \sqrt{x^T S^T S x} = \lVert Sx\lVert_2$
>
> **Ques 8)** Regarding.  *Line 180-181. "It means that the adjacency matrix A~ has a substantially smaller number of non-zero elements .....*
>
> **Ans 8)** As mentioned in the Line 177 coarsened graph, adjacency ($A_c$) is directly calculated from *$C^T A C$*. In general, $A_c$ has a substantially smaller number of non-zero elements than A. However, such extreme cases may exist where the number of edges remains the same and we will make a mention of such cases in the text if given an opportunity.
>
> **Ques 9)** Regarding *Where is the introduction of baselines .....*
>
> **Ans 9)** Section 2 includes the introduction to baseline(line 111-122)
>
> * UGC: Universal Graph Coarsening, our proposed method
> * VAN: Variation Neighborhood
> * VAE: Variation Edge
> * VC:  Variation Clique
> * HE:  Heavy Edge
> * aJC: Algebraic Distance
> * aGS: Affinity
> * Kron: Kron
>
> **Ques 10)** Regarding *Figure 7 is confusing. Present actual time saved’.....*
>
> **Ans 10)** The reason this figure is being presented is because the downstream tasks take substantially longer time as compared to the coarsening time in our case, whereas it is not true for the other methods. Adding a large number (downstream time) to a smaller one (coarsening time) could diminish the emphasis on coarsening, which is the focus of this work.

---

> > ### Comment · Reviewer_QBq8 · 2024-08-12
> >
> > I have read your replies and the replies corresponding to those of other fellow reviewers. My questions Q1, Q2, Q3, Q4, Q5, Q7, Q8, Q9, Q10 are settled; Q6 remains a shortcoming of the proposed method for the need of fixed node ordering, which should not be the concern of a `Graph model'; Consider the progresses been made, I recommend to consider `Borderline Accept' this paper where reasons to accept outweigh reasons to reject.

---

### Official Review · Reviewer_mriD · 2024-07-11

**Soundness:** 3
**Presentation:** 3
**Contribution:** 3
**Rating:** 6
**Confidence:** 4

**Summary:**

This paper proposes a new Universal Graph Coarsening (UGC) framework designed to handle both homophilic and heterophilic graphs. The UGC framework is capable of retaining important spectral properties, including eigenvalue error, hyperbolic error, and 𝜖-similarity measure. Experimental results demonstrate significant improvements in both performance and effectiveness with the UGC framework.

**Strengths:**

* Derive the coarsening matrix over graphs with heterophily property is an interesting idea, and Locality Sensitive Hashing (LSH) technique significantly reduces the algorithm's complexity.
* UGC offers significant speed improvements and is capable of handling both homogeneous and heterogeneous graphs.
* The experiments has demonstrated UGC’s limitations and possibility to improve the effectiveness across datasets.
* The paper is also very clear with thorough experiments and analysis.

**Weaknesses:**

1. Although the LSH strategy is much faster than other methods, the memory space overhead is non-negligible. Could you please measure the space complexity and provide a further empirical evaluation of the method? Additionally, how is the number of hash functions (𝑙) configured?
2. The paper lacks a detailed description of the experimental setup, including software and hardware environments, as well as the parameters used for different algorithms.
3. From Figure 5, we can see that UGC does not exhibit lower errors (RRE and HE) at lower Coarsening Ratios. I think it questions the claimed advantage of preserving spectral properties. Can the authors give a detail explanation?
4. Figure 5 illustrates that the methods (e.g., VAE, VC) do not maintain a monotonic relationship between the Coarsening Ratio and RRE/HE Error. Additionally, UGC does not exhibit a clear advantage across the overall Coarsening Ratio. Can the authors make a further clarify?
5. Table 3 does not prove the claim of “UGC (features) method achieves substantial accuracy enhancements” over GCN, GraphSAGE, GIN and GAT. Can the authors utilize UGC to more recent models, e.g., 3WL-GNNs, heterogeneous graph neural networks?
6. Minor issues：
    * In Figure 2, the specific meanings of a, b, and c are not labeled in the diagram.

**Questions:**

Please see the weakness section.

**Limitations:**

The authors have adequately addressed the limitation of the potential negative societal impact of their work.

---

> ### Author Rebuttal · Authors · 2024-08-07
>
> We thank the reviewer for their valuable comments and insights and for taking the time to go through our paper.
>
> **Ques 1)** *Although the LSH strategy is much faster than other methods, the memory space overhead is non-negligible. Could you please measure the space complexity and provide a further empirical evaluation of the method? Additionally, how is the number of hash functions (𝑙) configured?*
>
> **Ans 1)** The memory space overhead by UGC arises from two steps
>
> a) Random sampling of *$l \in R^d$* different projectors,
>
> b) Storing the hash values of each of 'n' nodes across these *l* projectors.
>
> Hence, the additional space complexity is bounded by O(l\*d + n\*l). *l* is a hyperparameter, for all our experiments, and all datasets, we have used 3000 different projectors(*l*). It is worth noting that for some datasets, the value of *l* can be as low as 1000.
>
> Additionally, due to its construction, UGC is suitable for online or streaming data as it does not require the entire feature matrix to be present simultaneously. In scenarios with limited memory space, the feature matrix can be split into different chunks, and UGC can be applied to each chunk separately, ensuring that the same memory is reused for all chunks. In this case, the required additional space complexity is bounded by O(l\*d + n'\*l) where n' is the size of the chunk and it can vary from 1 to n.
>
> **Ques 2)** *The paper lacks a detailed description of the experimental setup, including software and hardware environments, as well as the parameters used for different algorithms.*
>
> **Ans 2)** The details are included in the Appendix due to limited space in the main manuscript. For convenience, we also provide the details here:
>
> Appendix H contains a detailed discussion about the experimental setup as mentioned in Line 301. Figure 9 provides an overview of the GCN training pipeline, while Table 7 includes information on the hyper-parameters used for training.
>
> Appendix F contains details about the hardware specifications.
>
> "All experiments conducted for this work were performed on a desktop with an Intel Xeon W-295 CPU and 64GB of RAM using the Python environment."
>
> **Ques 3) and Ques4)**
> * *From Figure 5, we can see that UGC does not exhibit lower errors (RRE and HE) at lower Coarsening Ratios. I think it questions the claimed advantage of preserving spectral properties. Can the authors give a detail explanation?*
>
> * *Figure 5 illustrates that the methods (e.g., VAE, VC) do not maintain a monotonic relationship between the Coarsening Ratio and RRE/HE Error. Additionally, UGC does not exhibit a clear advantage across the overall Coarsening Ratio. Can the authors make a further clarify?*
>
> **Ans 3) and Ans4)** In Figure 5, lower coarsening ratios indicate that the graph is significantly reduced. We observe that the RRE error for UGC is optimal when the graph is reduced by approximately 0-65%. For HE error, we acknowledge that UGC is not the best-performing algorithm, but it is comparable to existing methods. However, for the node classification task, we can see that UGC gives the best results. How these properties are related to downstream tasks is not well understood yet and warrants further investigation.
>
> We have noticed the monotonic relationship that the reviewer mentioned but at the moment we are trying to analyze the data from additional experiments so that we can make an objective statement after a thorough assessment.
>
>
> **Ques 5)** *Table 3 does not prove the claim of “UGC (features) method achieves substantial accuracy enhancements” over GCN, GraphSAGE, GIN and GAT. Can the authors utilize UGC to more recent models, e.g., 3WL-GNNs, heterogeneous graph neural networks?*
>
> **Ans 5)**
>
> We have added the results of 3WL-GNN methods. It is to be noted that originally 3WL-GNN was a graph classification method, we have made the necessary changes to adapt it to node classification task.
>
>
> Data\Model | Var.Neigh | Var.Edges | Var.Clique | Heavy Edge | Alg. Dis. | Aff. GS | Kron | UGC | Full Dataset |
> |-|-|-|-|-|-|-|-|-|-|
> |Cora|**57.55**|43.30|53.12|60.31|60.01|52.93|56.75|55.32|63.45|
> |DBLP|48.60|49.83|51.76|51.95|51.74|52.13|52.19|**53.10**|61.58|
> |Physics|85.99|84.83|87.02|83.49|82.12|85.89|87.88|**88.56**|92.87|
> |Pubmed|69.94|64.29|70.42|63.11|53.55|27.04|62.30|**84.15**|86.68|
> |Squirrel|19.83|19.28|19.71|20.03|20.86|20.04|20.82|**61.92**|31.73|
> |Film|17.13|26.10|10.93|22.72|26.33|**27.16**|18.31|24.82|31.64|
> |Chameleon|18.40|22.59|20.03|23.78|23.04|16.63|23.24|**69.01**|44.61|
>
> In 5 out of 7 datasets UGC gives the best results for 3WL-GNN.
>
> **Ques 6)** *In Figure 2, the specific meanings of a, b, and c are not labeled in the diagram.*
>
> **Ans 6)** We thank the reviewer for pointing it out, Given the opportunity, we will improve the diagram to avoid any confusion.

---

> > ### Comment · Reviewer_mriD · 2024-08-12
> >
> > Thank you very much for your reply. My assessment of the paper remains positive.

---

### Official Review · Reviewer_C55f · 2024-07-14

**Soundness:** 3
**Presentation:** 3
**Contribution:** 2
**Rating:** 6
**Confidence:** 3

**Summary:**

The authors propose a novel Universal Graph Coarsening (UGC) framework, which is suitable for both homophilic and heterophilic datasets. UGC integrates node attributes and adjacency information to leverage dataset heterogeneity effectively. The results demonstrate that UGC is significantly faster (4x to 15x), maintains spectral similarity, and outperforms existing methods in terms of computational efficiency, eigen-error, and downstream processing tasks, especially at 70% coarsening ratios. The key contributions highlight UGC's universal applicability, efficiency, and information preservation.

**Strengths:**

1. The approach is intuitive and easy to understand.
2. The approach has a lower computational cost than most common method (Var.Neigh. etc,.)

**Weaknesses:**

For ScalableTrainingof Graph Neural Networks section, there are no detailed discussion on GNN models except GCN.

**Questions:**

For Table 4, why only GCN results are shown? I also expected results for GIN GAT and GraphSage.

**Limitations:**

Limitation is discussed in detail in papers.

---

> ### Author Rebuttal · Authors · 2024-08-07
>
> We thank the reviewer for their valuable comments and insights and for taking the time to go through our paper.
>
> **Ques 1)** *For ScalableTrainingof Graph Neural Networks section, there are no detailed discussion on GNN models except GCN.*
>
> **Ans 1)** Due to the limited space of the manuscript, we have only added a discussion about GCN to the manuscript. If the reviewer suggests, the following detailed discussion can be added to the paper:
>
> *GraphSAGE [1] is a scalable inductive framework for generating node embeddings in graphs. It leverages a neighborhood sampling and aggregation approach, allowing it to generalize to unseen nodes. This makes GraphSAGE particularly effective for large-scale graphs where retraining the model for new nodes would be computationally prohibitive. The Graph Isomorphism Network (GIN) [2] takes a different approach, designed to be as powerful as the Weisfeiler-Lehman graph isomorphism test. GIN uses a sum aggregation function, ensuring that different graph structures produce distinct embeddings. This ability to distinguish graph structures makes GIN a robust choice for tasks requiring high discriminative power. Graph Attention Networks (GAT) [3], on the other hand, introduce attention mechanisms to graph neural networks. GATs assign different importance to nodes in a neighborhood, which enhances the model's capability to focus on the most relevant parts of the graph. This attention mechanism allows GATs to achieve state-of-the-art performance on various node classification tasks by effectively capturing the underlying structure of the graph.*
>
> [1] Hamilton, Will, Zhitao Ying, and Jure Leskovec. "Inductive representation learning on large graphs." Advances in neural information processing systems 30 (2017).
>
> [2] Xu, K., Hu, W., Leskovec, J., & Jegelka, S. (2018). How powerful are graph neural networks?. arXiv preprint arXiv:1810.00826.
>
> [3] Veličković, Petar, et al. "Graph attention networks." arXiv preprint arXiv:1710.10903 (2017).
>
> **Ques 2)** *For Table 4, why only GCN results are shown? I also expected results for GIN GAT and GraphSage.*
>
> **Ans 2)** Due to space limitations in the manuscript, we included only GCN results in the main manuscript. However, we had already conducted experiments with GraphSage, GIN, and GAT for two homophilic and two heterophilic datasets, as shown in Table 3, to demonstrate the model-agnostic behavior of UGC.
>
> As suggested by the reviewer, we have now conducted experiments using GIN, GAT, and GraphSage models for Table 4. These results further demonstrate that UGC is not restricted to any specific model.
>
> | Dataset | Model | Var.Neigh | Var.Edges | Var.Clique | Heavy Edge | Alg. Dis. | Aff. GS | Kron | UGC |
> |-|-|-|-|-|-|-|-|-|-|
> ||||||||||||
> |    | gcn | 20.03 | 29.95 | 31.92 | 33.3 | 28.81 |27.58| 29.10| **48.7** |
> | Cham. | graphSage | 20.03| 20.02 | 22.05 | 23.03|19.88  |20.02| 27.62|**58.86** |
> | | gin |20.22 | 19.53| 25.25 |19.98  | 18.20   |   18.06 |  21.50 | **54.92** |
> | | gat |22.94|19.33 |26.44| 21.95 |23.72 | 18.06  | 21.95 | **55.58** |
> ||||||||||||
> |           | gcn | 19.67 | 20.22 | 19.54 | 20.36 | 19.96 |20.00|18.03 |**31.62** |
> | Squ. | graphSage | 19.87| 20.00 | 20.03 | 20.03 | 19.93  |20.00|19.98 |**57.60** |
> | | gin |18.54 | 19.65 | 18.98 | 21.65 |19.47 | 18.29   |20.56   | **35.64** |
> | | gat |20.90| 18.56|20.68| 19.93 |20.46 | 20.05  | 20.08 |**32.28**  |
> ||||||||||||
> |    | gcn | 15.67 | 21.80 |20.35  |19.16 |19.23| 20.34|17.41 | **25.40** |
> | Film | graphSage |22.32|26.05  |24.01 |21.49 | 21.88 |21.50|**23.73** |21.12 |
> | | gin |**24.20** |23.51  |17.51  |11.49  | 13.90|21.93 |18.04   |21.12 |
> | | gat |17.50|21.73 |17.82| 21.18 |17.94 | 17.40  |**24.15**  |21.71 |
> ||||||||||||||
> |    | gcn | 77.87 | 78.34 |73.32  | 74.66 | 74.59 |80.53|74.89 |**84.77**|
> | pubmed | graphSage |78.85 | 62.73 |67.18 |60.11 |63.09  |71.25|62.00 |**83.76** |
> | | gin |74.77 | 39.29 | 46.19 |35.97  | 32.13  | 49.63 | 39.29  | **76.36** |
> | | gat |75.22|72.63 | 74.81|60.04  |69.47| 59.76  | 71.92 |**83.56** |
> ||||||||||||
> |    | gcn |93.74  |93.86|92.94|93.03|93.94|93.06|92.26 |**96.12** |
> | physics | graphSage |OOM |OOM|OOM |OOM |OOM|OOM|OOM |OOM |
> | | gin |OOM |OOM |OOM  |OOM  |OOM |OOM|OOM| OOM |
> | | gat |92.04|91.80|91.48 | 91.80 | 92.94 |  93.33 | 91.60 | **93.80**  |
> ||||||||||||
> |    | gcn | 77.05 | **79.93** | 79.15 |77.46 | 74.51 |78.15|77.79 | 75.50  |
> | dblp | graphSage |68.54 |60.17  |**74.17** |72.70 | 72.19 |71.81| 71.76| 68.25 |
> | | gin |35.84 |33.93  |35.12  | 24.16 |  51.47  |  47.30  | 42.24  |**55.28**|
> | | gat |70.20|**74.07** |72.82| 71.35 | 71.17| 76.12  | 72.27 |73.49|
> ||||||||||||
> |    | gcn | 79.75 | 81.57 |80.92  |79.90 |79.83  |80.20|80.71 |**86.30** |
> | cora | graphSage |70.49 | 68.48 |70.16 |69.17 | 72.26 |67.77|**73.20** | 69.39 |
> | | gin |47.65 | 35.03 | 52.91 |34.00  |  63.05  | 23.49  | 48.56  | **67.23**  |
> | | gat |69.26|74.02 |**75.92**| 68.95 |73.09 | 73.83  |73.24 | 74.21 |
> ||||||||||||
>
> If the reviewer suggests, we can include these additional results in the Appendix and refer to them in the caption of Table 4 as follows:
>
> Table 4: This table illustrates the accuracy of the GCN model when trained with a 50% coarsened graph. UGC demonstrated superior performance compared to existing methods in 7 out of the 9 datasets. Please refer to Appendix H for results with GraphSage, GIN, and GAT models.

---

> > ### Author Response · Authors · 2024-08-13
> > **Eagerly awaiting feedback on rebuttal**
> >
> > Dear Reviewer,
> >
> > Since we are only a day away from the completion of the discussion phase, we are eagerly awaiting your feedback on the rebuttal.
> >
> > Your review pointed out important empirical studies that further enhanced our work. We have incorporated all of them and we thank the reviewer again for the deep insightful comments on our work. We would love to discuss more if any concern remains unaddressed. Otherwise, we would really appreciate it if you could support the paper by increasing the score.
> >
> > regards,
> >
> > Authors

---

> > ### Comment · Reviewer_C55f · 2024-08-13
> >
> > Thanks for your reply. It resolved my concern, and I decided to increase my rating.

---

### Official Review · Reviewer_FD1i · 2024-07-15

**Soundness:** 3
**Presentation:** 3
**Contribution:** 3
**Rating:** 7
**Confidence:** 4

**Summary:**

This paper present a framework UGC for graph coarsening to reduce a larger graph to a smaller graph. It uses Locality Sensitive Hashing (LSH) of augmented node features, and works on both homophily and heterophilic graphs. Experiments could verify its effectiveness in original graph property perservation and efficiency in coarsening speed.

**Strengths:**

S1: This work proposes to use hashing method to graph coarsening and works well on universal homophilic and heterophilic graphs, which is interesting and rational to me.


S2: The overall logic, problem definition, and the solution, are well described and clearly illustrated. The presentation is good and not hard to follow.

S3: UGC is faster compared to existing methods. It achieves a reduction in graph size with lower computational time, making it suitable for large datasets.

**Weaknesses:**

Here are some questions need to be further addressed:

W1: In lines 149-150, it claims that the augmented feature vector is calculated by dot product and concatenation, it is not very clear that how to concate A with node features X?


W2: Some figures, e.g., figure 1 and figure 4, is not very clear for visualization.

W3: In table 4, for comparisons in GNN classification accuracy, it would be better to involve some other graph size reduction methods, like GCOND, SFGC to show the performance.

W4: Is there any ablation study to show the effectiveness of the proposed augmented feature vector?

W5: More detailed explanations and analysis of whether the proposed method could adapt heterophilic graph well are expected. Is there any specific design targeting for heterophily?

**Questions:**

See Weaknesses.

**Limitations:**

Yes.

---

> ### Author Rebuttal · Authors · 2024-08-07
>
> We thank the reviewer for their valuable comments and insights and for taking the time to go through our paper.
>
> **Ques 1)** *it claims that the augmented feature vector is calculated by dot product and concatenation, it is not very clear that how to concate A with node features X*
>
> **Ans 1)**  We thank the reviewer for bringing this typo to our attention.  The augmented feature vector is *calculated by scaling and concatenation operation* instead of *dot product and concatenation*.
>
> The augmented feature vector of node *$v_i$* is calculated by scaling the adjacency vector *$A_i$* by ($\alpha$) and feature vector *$X_i$* by *(1 - $\alpha$)*, followed by the concatenation operations of the two vectors as (1 - $\alpha$)*$X_i$ || ($\alpha$) * $A_i$, where $\alpha$ is the heterophily factor.
>
> A detailed illustration of this process is given in Figure 11 in Appendix K. This figure provides a toy example demonstrating how the augmentation matrix is formulated. It is also mentioned in the main manuscript, specifically on lines 150-151.
>
> **Ques 2)** *Some figures, e.g., figure 1 and figure 4, is not very clear for visualization.*
>
> **Ans 2)** Thank you for bringing this to our attention. We will address this issue by increasing the size of the axis labels in Figure 1 and Figure 4 to enhance their clarity. We appreciate your feedback and will make these adjustments in the updated manuscript.
>
> **Ques 3)**  *In table 4, for comparisons in GNN classification accuracy, it would be better to involve some other graph size reduction methods, like GCOND, SFGC to show the performance.*
>
> **Ans 3)** We thank the reviewer for the suggestion. We have added the accuracy of GCond in the following table.
>
> GCond accuracy and time
> Data | GCond Accuracy | GCond Coarsening Time |GCN training time on original graph|UGC Coarsening Time(x Fast compared to GCond) |UGC accuracy|
> |-------|-------------|---------------|--|---------------|--|
> |Cora|80.43|2640|25.77|0.41(x6440)|86.30|
> |Pubmed|76.98|1620|114.55|1.62(x1000)|84.77|
> |Physics|OOM|-|1195.56|6.4|96.12|
> |DBLP|82.63|25500|174.10|1.86(x13710)|75.50|
> |Squirrel|59.64|7860|228.52|2.14(x3673)|31.62|
> |Chameleon|52.29|7740|54.34|0.49(x15796)|48.7|
>
> As mentioned in the paper, these methods are computationally demanding, which is also evident from the table above. Specifically, the time required to coarsen the graph exceeds the time needed to train the GNN on the original graphs.
>
> These results can be included in Table 1 and Table 4 if suggested by the reviewer.
>
> **Ques 4)** *Is there any ablation study to show the effectiveness of the proposed augmented feature vector?*
>
> **Ans 4)** Yes, the ablation study to demonstrate the effectiveness of the proposed augmented feature vector is included in Table 4. In this table, "UGC feat." denotes the scenario where $\alpha$ is set to zero, meaning only the feature matrix is considered, while "UGC-feat. + adj." denotes the case where $\alpha$ is set to heterophily factor, thereby incorporating the adjacency vector.
>
> For heterophilic datasets, node classification accuracy improves significantly when using the augmented feature vector. This highlights the importance of the adjacency vector in the augmented feature vector.
>
>
> **Ques 5)** *More detailed explanations and analysis of whether the proposed method could adapt heterophilic graph well are expected. Is there any specific design targeting for heterophily?*
>
> **Ans 5)** We thank the reviewer for bringing this up. We have observed that the heterophily factor can be directly utilized as the $\alpha$ value. When we extrapolated the results for different $\alpha$ values, it was observed that setting $\alpha$ around the heterophily factor yielded the best results.
>
> The results are shown in the table below for two heterophilic datasets, Squirrel and Chameleon. For both datasets, the heterophily factor is approximately 0.78. We observed that the best results for these datasets are obtained when $\alpha$ is set around heterophily factor.
>
>
> | $\alpha$ value | GCN accuracy for Squirrel| GCN accuracy for Chameleon|
> |-------------|---------------|-|
> |0|20.71|29.90|
> |0.1|24.14|38.02|
> |0.2|26.71|42.86|
> |0.3|27.03|43.74|
> |0.4|28.12|41.98|
> |0.5|27.89|40.00|
> |0.6|28.93|47.91|
> |0.7|29.82|**49.49**|
> |0.8|**31.62**|49.45|
> |0.9|29.93|46.59|
> |1.0|28.46|46.15|

---

> > ### Author Response · Authors · 2024-08-13
> > **Looking forward to your feedback on rebuttal**
> >
> > Dear Reviewer,
> >
> > We thank you for the insightful comments on our work. Your suggestions have now been incorporated in our revision and we are eagerly waiting for your feedback. As the author-reviewer discussion phase is approaching its conclusion in just a few hours, we are reaching out to inquire if there are any remaining concerns or points that require clarification. Your feedback is crucial to ensure the completeness and quality of our work.
> >
> > We are pleased to share that the responses from other reviewers also indicate a positive inclination toward acceptance. Your support in this final phase, would be immensely appreciated.
> >
> > regards,
> >
> > Authors

---

### Author Rebuttal · Authors · 2024-08-07

We thank the reviewers for their insights and constructive suggestions. A comprehensive point-by-point response to the reviewers' comments is presented below. The major additional changes are listed below.

**Additional experiments**: We have incorporated all of the additional experiments requested by the reviewers spanning

* Adding the GCond node classification accuracies and computational time in the rebuttal.
* Extending Table 4 results to include GraphSage, GIN, and GAT models.
* Conducting experiments with varying values of the $\alpha$ hyperparameter from [0,1] to justify the UGC design for handling heterophily datasets.
* Including node classification experiments with the 3WL-GNNs model.

We hope these revisions will satisfactorily address the concerns raised by the reviewers and elevate the overall quality of our work.

---

### Author Response · Authors · 2024-08-10
**Looking forward to your feedback on rebuttal**

Dear Reviewers,

Thank you once again for all of your constructive comments, which have helped us significantly improve the paper! As detailed below, we have performed several additional experiments and analyses to address the comments and concerns raised by the reviewers.

Since we are into the last two days of the discussion phase, we are eagerly looking forward to your post-rebuttal responses.

Please do let us know if there are any additional clarifications or experiments that we can offer. We would love to discuss more if any concern still remains. Otherwise, we would appreciate it if you could support the paper by increasing the score.

Thank you!

Authors

---

### Decision · Program_Chairs · 2024-09-25

**Decision:**

Accept (poster)

**Comment:**

This paper present a framework for graph coarsening to reduce a larger graph to a smaller graph that works on both homophilic and heterophilic graphs. Empirical results verify the effectiveness of the method in preserving graph properties as well as in in coarsening speed.